# Reactivation during sleep with incomplete reminder cues rather than complete ones stabilizes long-term memory in humans

Cecilia Forcato [1,2 ✉], Jens G. Klinzing[3,4,5], Julia Carbone[3], Michael Radloff[3], Frederik D. Weber [6], Jan Born [3,4] & Susanne Diekelmann[3]

Reactivation by reminder cues labilizes memories during wakefulness, requiring reconsolidation to persist. In contrast, during sleep, cued reactivation seems to directly stabilize memories. In reconsolidation, incomplete reminders are more effective in reactivating memories than complete reminders by inducing a mismatch, i.e. a discrepancy between expected and actual events. Whether mismatch is likewise detected during sleep is unclear. Here we test whether cued reactivation during sleep is more effective for mismatch-inducing incomplete than complete reminders. We first establish that only incomplete but not complete reminders labilize memories during wakefulness. When complete or incomplete reminders are presented during 40-min sleep, both reminders are equally effective in stabilizing memories. However, when extending the retention interval for another 7 hours (following 40-min sleep), only incomplete but not complete reminders stabilize memories, regardless of the extension containing wakefulness or sleep. We propose that, during sleep, only incomplete reminders initiate long-term memory stabilization via mismatch detection.

[1] Laboratorio de Sueño y Memoria, Depto. de Ciencias de la Vida, Instituto Tecnológico de Buenos Aires (ITBA), Av. Madero 399, 1106 Capital Federal, Buenos Aires, Argentina. [2] Consejo Nacional de Investigaciones Científicas y Tecnológicas (CONICET), Godoy Cruz 2290, 1425 Capital Federal, Buenos Aires, Argentina. [3] Institute of Medical Psychology and Behavioral Neurobiology, University of Tübingen, Otfried-Müller-Straße 25, 72076 Tübingen, Germany. [4] Centre for Integrative Neuroscience (CIN), University of Tübingen, Otfried-Müller-Straße 25, 72076 Tübingen, Germany. [5] Princeton Neuroscience Institute, Princeton University, Washington Rd, Princeton, NJ 08540, USA. [6] Donders Institute for Brain, Cognition and Behaviour, Radboudumc, Kapittelweg 29, 6525 EN Nijmegen, The Netherlands. ✉email: cforcato@itba.edu.ar

  1

The formation and modification of long-term memories represent core capabilities of organisms to adapt to complex environments. The reactivation of a memory by presenting a reminder cue in the wake state can return already consolidated memories to a labile state followed by a period of restabilization known as reconsolidation[1–3]. This labilization/reconsolidation process serves two important functions: the updating of old memories with new information and the strengthening of the original memories[4–6]. Both functions depend on a prediction error induced by the reminder cue, that is, a mismatch between expected and current events[7–10]. Such a mismatch is most readily induced by incomplete reminder cues that include only parts of the original learning information, for example when an expected reinforcement after the presentation of the cue is omitted (e.g. single cue syllable), as opposed to reminders formed by the complete stimulus association (e.g. pairs of syllables in a syllable pair-association task)[4,8,11–14]. Incomplete reminders were more effective than complete reminders in triggering the reactivation and updating of memories when single incomplete reminders were presented in the wake state, followed by interference learning within the labilization window of ~6 h[5,15]. Incomplete reminder cues were also more effective in inducing memory strengthening when, instead of interference learning, a second incomplete reminder was presented while the memory was still in a labile state[6,16]. However, this strengthening effect was not observed when a complete reminder was presented in this time window of lability[6].

Reactivation of memory representations also occurs spontaneously during sleep following a learning experience, a process known as replay[17]. Especially during non-rapid eye movement (NREM) sleep, neuronal networks involved in the encoding of recent experiences express patterns of activation similar to those observed during learning[17–26]. Memory reactivation can also be experimentally induced during sleep through external stimulation by reminder cues[27], with this cued reactivation distinctly facilitating the targeted memories[28–37]. In contrast to reactivation during wakefulness, cued reactivation during sleep was found to stabilize memories immediately. In a study applying learning-associated odor cues, re-exposure to the odor during 40 min of sleep improved resistance against interference, with memories being less susceptible to disruption by interference learning during subsequent wakefulness[38]. Despite compelling evidence for the beneficial effects of cued memory reactivation during sleep, there are only few studies on the mechanisms underlying memory stabilization upon cued reactivation in the sleep state. These studies show that hippocampal replay is linked to cued reactivation[27] and that cortical input during reactivation is part of a cortico–hippocampal–cortical loop, strengthening memory traces through the reverberation of replay between the cortex and hippocampus[39].

Interestingly, previous sleep cueing studies almost exclusively applied cues that can be classified as incomplete reminders[40–42]. For instance, odor cues can be considered incomplete reminders because they serve as context cues for the associated learning material[33,38]. Sound cues are typically associated with other contents like words or pictures and can therefore be considered incomplete reminders as well. Memory strengthening in sleep reactivation studies is frequently observed after only a single presentation of the incomplete reminder, while in wake reactivation studies a second incomplete reminder is necessary to induce memory strengthening after the first incomplete reminder has triggered labilization. This difference could be explained by the fact that reminder presentation in sleep studies almost always takes place shortly after learning, i.e., while the memory is still in a labile state and not already consolidated as in typical wake reactivation studies. However, it is presently unclear whether cueing with incomplete reminders is more effective than cueing with complete reminders during sleep, similar to the wake state. Memory consolidation during sleep has recently been conceptualized as a complex process involving the restructuring and integration of new contents into pre-existing networks, including processes of abstraction and schema extraction[43,44]. For such an integrative process, labilization and updating may be essential for a complex re-configuration of the whole network. Considering that labilization and updating during wakefulness depend on incomplete reminder presentation, we expect cueing with incomplete reminders during sleep to be more effective than with complete reminders. Some evidence suggests that at least a rudimentary form of mismatch detection, which is assumed to underlie incomplete reminder effects during wakefulness, is still in place during sleep[45].

Based on these considerations, here we conducted two separate studies. The first study examined, in a new sound-word association paradigm, whether incomplete reminders are more effective than complete reminders in inducing labilization and memory updating in a classical 3-day reconsolidation paradigm during wakefulness. The second study compared cued memory reactivation during sleep with incomplete and complete reminders in three experiments, varying the time interval between cueing and testing to examine memory stabilization (i.e. resistance against interference) on the short-term and on the long-term.

The results of the first study show that only incomplete reminder cues but not complete ones induce memory labilization/reconsolidation during wakefulness. The second study revealed that during sleep, both incomplete and complete reminders stabilize memories on the short term, whereas only incomplete reminders initiate long-term memory stabilization, regardless whether the longer retention interval contains additional wakefulness or sleep.

## Results

**Study 1 – reconsolidation during wakefulness.** We first examined whether incomplete or complete reminders trigger memory reconsolidation in a new sound-word association paradigm (Fig. 1). Participants learned 30 sound-word associations on Day 1. On Day 2, they were reactivated with either complete reminders (sounds plus associated words) or incomplete reminders (sounds plus first syllables of the associated words), followed or not by an interference learning task to impair memory restabilization. The "complete reminder/no interference" group received the complete reminder without the interference task; the "complete reminder/interference" group received the complete reminder followed by the interference task; the "incomplete reminder/no interference" group received the incomplete reminder without the interference task; the "incomplete reminder/interference group" received the incomplete reminder followed by the interference task. Participants were tested on Day 3.

Initial learning performance on Day 1 was similar between groups (two-way ANOVA, reminder type: $F_{1,44} = 0.28$, $P = 0.60$, $\eta^2 = 0.006$; interference: $F_{1,44} = 0.12$, $P = 0.91$, $\eta^2 = 0.000$; interaction: $F_{1,44} = 0.63$, $P = 0.80$, $\eta^2 = 0.001$, Table 1). However, striking differences developed for the memory change from Day 1 to Day 3 (Fig. 2b), with the interference task exerting differential effects depending on the type of reminder received on Day 2 (two-way ANOVA, interaction: $F_{1,44} = 6.92$, $P = 0.012$, $\eta^2 = 0.136$; reminder type: $F_{1,44} = 7.11$, $P = 0.01$, $\eta^2 = 0.139$; interference: $F_{1,44} = 1.76$, $P = 0.19$, $\eta^2 = 0.038$). For the complete reminder groups, performance was comparable independent of whether or not they received interference after reactivation (no

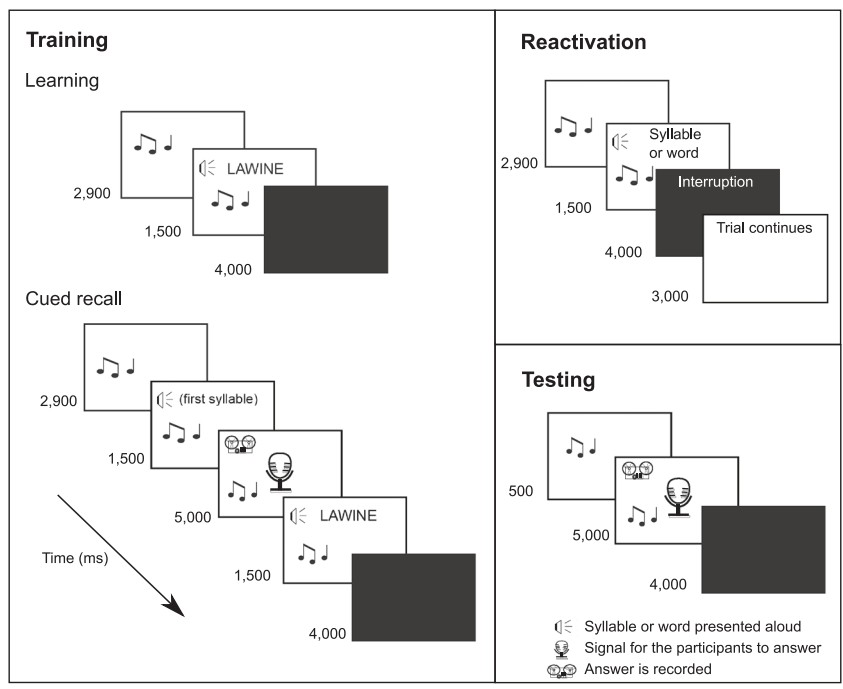

**Fig. 1 Memory task.** During the training session, participants learned 30 sound-word associations. For each association, the sound was presented first followed by the presentation of the sound plus the word written on the screen and spoken aloud via headphones. The next association appeared after a 4000-ms break. After all 30 associations had been presented once (learning), participants completed an immediate cued recall test. For each association, the sound and the first syllable of the associated word were presented and participants had to say the associated word aloud upon appearance of an image of a microphone on the screen, while the sound continued during the entire period. Independent of the participant's response, the correct answer was then displayed written on the screen and via headphones. In the reactivation session, each sound was first presented alone for on average 2900 ms. The sound then continued and the first syllable of each word (incomplete reminder) or the entire word (complete reminder) was presented once in addition to the sound for 1500 ms. Although in Study 1, participants were instructed to say the associated word aloud every time the microphone appeared on the screen, the microphone never actually appeared and each trial was interrupted so that participants never had the chance to give a response. In Study 2, exactly the same reactivation procedure was applied during sleep, except for the instruction to respond. In the testing session, participants were presented with the sound of each association and had to say the associated word aloud. Musical notes indicate presentation of the sound.

**TABLE 1 Memory task results and control measures in Study 1.**

|  | Rc/no interference | Rc/ interference | Ri/no interference | Ri/ interference |
|---|---|---|---|---|
| *Memory task* |  |  |  |  |
| Training | 21.0 ± 1.3 | 21.5 ± 1.5 | 20.6 ± 1.1 | 20.4 ± 1.7 |
| Interference | – | 24.7 ± 1.3 | – | 23.6 ± 1.3 |
| Testing | 22.7 ± 1.5 | 24.5 ± 1.6 | 22.3 ± 1.2 | 18.3 ± 2.1 |
| *SSS* |  |  |  |  |
| Training | 2.3 ± 0.3 | 2.2 ± 0.2 | 1.9 ± 0.2 | 2.0 ± 0.2 |
| Interference | 2.5 ± 0.2 | 2.4 ± 0.3 | 2.2 ± 0.3 | 2.2 ± 0.2 |
| Testing | 2.2 ± 0.3 | 2.4 ± 0.2 | 1.9 ± 0.2 | 2.3 ± 0.3 |

Mean number of correct responses is indicated for the memory task ±SEM at training of the original task (Training), training of the interference task (Interference) and testing of the original task (Testing), as well as mean rating in the Stanford Sleepiness Scale (SSS) ± SEM for the four groups of Study 1.
*Rc* complete reminder, *Ri* incomplete reminder.

interference $+1.7 \pm 0.7$, interference $+3.0 \pm 0.8$; Simple effects, $F_{1,44} = 0.82$, $P = 0.37$, $\eta^2 = 0.018$). In contrast, the groups that received the incomplete reminder showed a different memory pattern depending on the presence or absence of the interference task. Specifically, learning the interference task after incomplete reminder presentation significantly impaired performance compared to the condition without interference learning (simple effects, no interference $+1.7 \pm 1.0$, interference $-2.2 \pm 1.3$; $F_{1,44} =$

8.17, $P = 0.006$, $\eta^2 = 0.157$). Furthermore, when subjects received the incomplete reminder before the interference task they performed significantly worse than subjects who received the complete reminder before interference learning (Simple effects, $F_{1,44} = 14.08$, $P < 0.001$, $\eta^2 = 0.242$), whereas both reminder groups that received no interference showed comparable performance levels (Simple effects, $F_{1,44} = 0.001$, $P = 0.98$, $\eta^2 = 0.000$). No difference was found between groups for learning of the interference task (one-way ANOVA, $F_{1,22} = 0.35$, $P = 0.56$, $\eta^2 = 0.015$, Table 1). Groups did also not differ in subjective sleepiness ratings at training (Two-way ANOVA, reminder type: $F_{1,44} = 1.22$, $P = 0.28$, $\eta^2 = 0.027$; interference: $F_{1,44} = 0.004$, $P = 0.95$, $\eta^2 = 0.000$; interaction: $F_{1,44} = 0.15$, $P = 0.70$, $\eta^2 = 0.003$, Table 1), interference learning (two-way ANOVA, reminder type: $F_{1,44} = 1.14$, $P = 0.29$, $\eta^2 = 0.025$; interference: $F_{1,44} = 0.002$, $P = 0.96$, $\eta^2 = 0.000$; interaction: $F_{1,44} = 0.010$, $P = 0.92$, $\eta^2 = 0.000$), and testing (two-way ANOVA, reminder type: $F_{1,44} = 0.71$, $P = 0.41$, $\eta^2 = 0.016$; interference: $F_{1,44} = 1.23$, $P = 0.27$, $\eta^2 = 0.027$; interaction: $F_{1,44} = 0.03$, $P = 0.86$, $\eta^2 = 0.001$).

These results show that during wakefulness only the incomplete reminder labilizes the sound-word associations making them vulnerable for disruption by the interference task, whereas the complete reminder does not labilize the underlying memory traces. Note that this study does not allow for any conclusions regarding memory strengthening, which would require comparing the reminder groups to a group without reminder presentation.

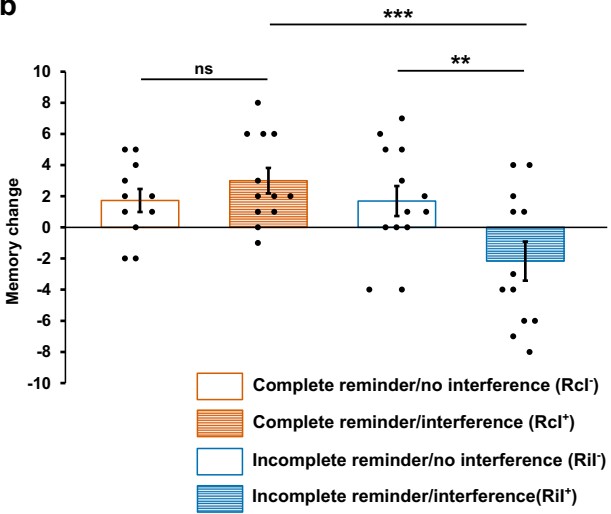

**Fig. 2 Memory labilization by different reminder cues during wakefulness in Study 1. a** Four groups of participants were examined in a 3-day reconsolidation paradigm. All participants were trained on Day 1. On Day 2, one group of participants received the complete reminder without any interference learning ($R_cI^-$), the second group received the complete reminder followed by interference learning ($R_cI^+$), the third group received the incomplete reminder without interference learning ($R_iI^-$), and the fourth group received the incomplete reminder followed by interference learning ($R_iI^+$). Testing of the original task took place on Day 3. Sample size: $R_cI^-$: $n = 11$; $R_cI^+$: $n = 12$; $R_iI^-$: $n = 13$; and $R_iI^+$: $n = 12$, independent samples. **b** Interference learning affected memory differentially depending on reminder type. While the incomplete reminder group was impaired in memory performance by interference learning, the complete reminder group remained unaffected. Memory change (y-axis) represents the number of correct responses at testing minus the number of correct responses at training, i.e., indicating how many items subjects gained or lost from training to testing. Means ± SEM are shown. The orange bar represents the "complete reminder/no interference" group; the orange striped bar represents the "complete reminder/interference" group; the blue bar represents the "incomplete reminder/no interference" group and the blue striped bar represents the "incomplete reminder/interference" group. ns not significant; **$P < 0.01$; ***$P < 0.001$.

## Study 2 – cued memory reactivation during sleep.

Study 2 was composed of three independent experiments (Exp. 2–4). We first examined the short-term effect of memory reactivation with different types of reminders during SWS (Exp. 2). Then, we looked at the long-term effect of memory reactivation with different types of reminders during SWS (Exp. 3). Finally, we tested whether the observed long-term effects depend on extended sleep or rather on the simple passage of time (Exp. 4). Note that in Exp. 2 and 4 the envisaged duration of the short sleep period was ~40 min as in Diekelmann et al.[38]. Although some participants in Exp. 2 and 4 slept somewhat longer than that (mean: 53.3 min, SD: 23.8, see Methods section for details), we stick to the term "40-min sleep" in the following for reasons of consistency.

## Experiment 2: short-term memory stabilization with complete vs. incomplete reminders during SWS.

Subjects were trained on Day 1 before going to sleep. During SWS they received either the complete reminder ("complete reminder/40 min" group), the incomplete reminder ("incomplete reminder/40 min" group) or they were not reactivated ("no reminder/40 min" group). After ~40 min of sleep they were awakened, learned the interference task and were tested on the original task (Fig. 3a).

Learning performance was comparable between groups (one-way ANOVA, $F_{2,30} = 0.30$, $P = 0.75$, $\eta^2 = 0.019$; Table 2). However, significant differences emerged for memory change from the learning session to the testing session (Fig. 3b, one-way ANOVA, $F_{2,30} = 5.65$, $P = 0.008$, $\eta^2 = 0.273$). Both reminder groups performed distinctly better than the group that did not receive any reminder during sleep ("no reminder/40 min" $-6.0 \pm 1.3$, "complete reminder/40 min" $-2.1 \pm 0.7$, and "incomplete reminder/40 min" $-1.4 \pm 1.1$). The difference in performance as compared with the no reminder group was significant for both the "complete reminder/40 min" group ($T_{20} = 2.65$, $P = 0.015$, $r = 0.51$) as well as the "incomplete reminder/40 min" group ($T_{20} = 2.79$, $P = 0.011$, $r = 0.53$). Performance of the "complete reminder/40 min" and the "incomplete reminder/40 min" groups did not differ ($T_{20} = 0.56$, $P = 0.58$, $r = 0.12$).

There was no difference between groups in learning of the interference task (one-way ANOVA, $F_{2,30} = 0.54$, $P = 0.59$, $\eta^2 = 0.034$; Table 2), nor in subjective sleepiness (one-way ANOVAs, at training: $F_{2,30} = 0.50$, $P = 0.61$, $\eta^2 = 0.032$, interference: $F_{2,30} = 0.99$, $P = 0.39$, $\eta^2 = 0.062$, testing: $F_{2,30} = 0.67$, $P = 0.52$, $\eta^2 = 0.043$; Table 2). In the "heard/not-heard" task, subjects correctly indicated only very few words and there was no significant difference between groups (one-way ANOVA, $F_{2,30} = 0.09$, $P = 0.91$, $\eta^2 = 0.006$; Table 2). With regard to sleep stage distribution, there were no significant differences for total time of sleep (Table 3, one-way ANOVA, $F_{2,30} = 0.55$, $P = 0.59$, $\eta^2 = 0.035$) as well as percentage of wake (one-way ANOVA, $F_{2,30} = 0.34$, $P = 0.71$, $\eta^2 = 0.022$), stage 1 (one-way ANOVA, $F_{2,30} = 0.59$, $P = 0.56$, $\eta^2 = 0.038$), stage 2 (one-way ANOVA, $F_{2,30} = 1.06$, $P = 0.36$, $\eta^2 = 0.066$) and SWS (one-way ANOVA, $F_{2,30} = 0.15$, $P = 0.86$, $\eta^2 = 0.001$).

These findings indicate that on the short-term, i.e. after a ~40-min sleep period, both incomplete and complete reminders trigger memory stabilization.

## Experiment 3: long-term memory stabilization with complete vs. incomplete reminders during SWS and additional sleep.

The procedures were identical to Experiment 2, except that after the ~40-min sleep period with reminder presentation, participants were allowed to continue sleeping for another 7 h without any additional reminders (altogether sleeping for ~8 h; Fig. 3a).

Like in Experiment 2, learning performance was comparable between the three groups (one-way ANOVA, $F_{2,39} = 1.18$, $P = 0.32$, $\eta^2 = 0.057$; Table 2). However, significant differences emerged for memory change from the learning session to the testing session (Fig. 3c, one-way ANOVA, $F_{2,39} = 3.38$, $P = 0.04$, $\eta^2 = 0.148$). In contrast to Experiment 2, the "incomplete reminder/8 h" group performed significantly better than both the "complete reminder/8 h" group ($T_{25} = 2.30$, $P = 0.03$, $r = 0.42$) and the "no reminder/8 h" group ($T_{26} = 2.18$, $P = 0.038$; $r = 0.39$) ("incomplete reminder/8 h" $-1.9 \pm 1.1$, "complete reminder/8 h" $-6.5 \pm 1.6$, "no reminder/8 h" $-4.9 \pm 0.9$). Performance of the "complete reminder/8 h" group and the "no reminder/8 h" group did not differ ($T_{27} = -0.85$, $P = 0.40$, $r = 0.16$).

Like in Experiment 2, there was no difference between groups in learning of the interference task (one-way ANOVA, $F_{2,39} = 0.82$, $P = 0.45$, $\eta^2 = 0.041$; Table 2), subjective sleepiness (one-way ANOVAs, at training: $F_{2,39} = 0.93$, $P = 0.40$, $\eta^2 = 0.046$; interference: $F_{2,39} = 0.06$, $P = 0.99$, $\eta^2 = 0.000$; testing:

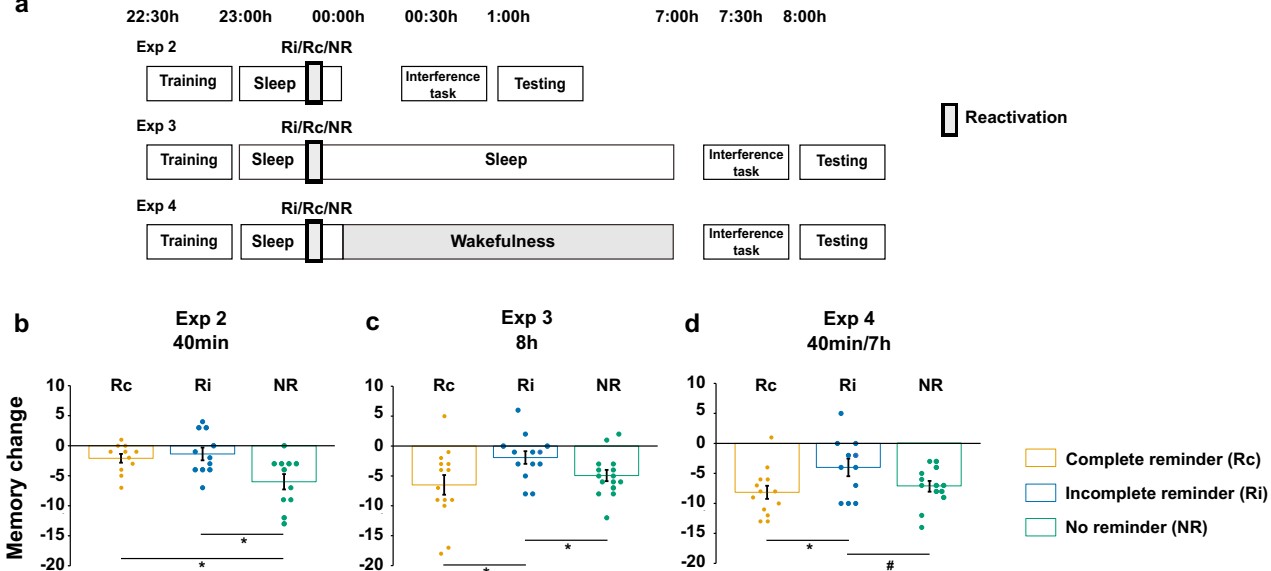

**Fig. 3 Memory stabilization by different reminder cues during sleep in Study 2. a** In three experiments, all participants were trained in the evening before a ~40-min sleep period during which the complete reminder (Rc groups), incomplete reminder (Ri groups), or no reminder (NR groups) was presented. In Exp. 2, participants were awakened after the ~40-min sleep period, learned the interference task 30 min later and were tested on the original task another 30 min later. In Exp. 3, procedures were exactly the same for training and reminder presentation, but participants were allowed to continue sleeping after reminder presentation and were awakened in the morning, with interference learning and testing taking place thereafter. In Exp. 4, procedures were also identical for training and reminder presentation, but participants were awakened after the ~40 min sleep period and remained awake for another 7 h, with interference learning and testing taking place thereafter. **b** In Exp. 2 (40 min), both reminder groups performed better than the no reminder group, with no difference between the complete and incomplete reminder groups. Sample size: all groups $n = 11$, independent samples. **c** In Exp. 3 (8 h), the incomplete reminder group performed better than the complete reminder and no reminder groups, with no difference between the complete and no reminder groups. Sample size: Rc: $n = 14$; Ri: $n = 13$; and NR: $n = 15$, independent samples. **d** In Exp. 4 (40 min-7h), the incomplete reminder group performed better than the complete reminder group and showed a trend toward better performance than the no reminder group, with no difference between the complete and no reminder groups. Sample size: Rc: $n = 13$; Ri: $n = 11$; and NR: $n = 13$, independent samples. Memory change represents the number of correct responses at testing minus training, i.e., indicating forgetting. Means ± SEM are shown. Orange bars represent the complete reminder groups, blue bars the incomplete reminder groups and green bars the no reminder groups. *$P < 0.05$; #$P = 0.072$.

**TABLE 2 Memory task results and control measures in Study 2.**

|  | Exp. 2 (40 min) | | | Exp. 3 (8 h) | | | Exp. 4 (40 min-7h) | | |
|---|---|---|---|---|---|---|---|---|---|
|  | Rc | Ri | NR | Rc | Ri | NR | Rc | Ri | NR |
| *Memory task* | | | | | | | | | |
| Training | 23.0 ± 1.3 | 22.5 ± 1.4 | 21.7 ± 0.9 | 23.1 ± 1.3 | 20.9 ± 1.0 | 23.1 ± 1.1 | 24.5 ± 1.0 | 23.6 ± 1.1 | 22.2 ± 1.1 |
| Interference | 24.4 ± 1.2 | 24.1 ± 0.9 | 22.8 ± 1.2 | 23.4 ± 1.0 | 23.8 ± 1.1 | 24.9 ± 0.7 | 22.7 ± 1.5 | 24.6 ± 1.3 | 21.6 ± 1.1 |
| Testing | 20.9 ± 1.3 | 21.1 ± 1.0 | 15.7 ± 1.7 | 16.6 ± 1.8 | 19.0 ± 1.2 | 18.2 ± 1.3 | 16.3 ± 1.8 | 19.6 ± 1.9 | 15.0 ± 1.5 |
| *SSS* | | | | | | | | | |
| Training | 2.6 ± 0.3 | 2.9 ± 0.3 | 3.1 ± 0.4 | 3.4 ± 0.4 | 3.2 ± 0.3 | 2.8 ± 0.2 | 3.1 ± 0.2 | 3.5 ± 0.3 | 3.0 ± 0.3 |
| Interference | 3.3 ± 0.3 | 4.1 ± 0.5 | 3.5 ± 0.5 | 2.6 ± 0.3 | 2.6 ± 0.2 | 2.6 ± 0.3 | 4.7 ± 0.4 | 5.6 ± 0.5 | 5.2 ± 0.4 |
| Testing | 3.8 ± 0.3 | 3.8 ± 0.3 | 3.4 ± 0.4 | 2.3 ± 0.2 | 2.4 ± 0.2 | 2.3 ± 0.3 | 4.6 ± 0.3 | 5.3 ± 0.4 | 5.2 ± 0.3 |
| *Heard/not-heard task* | | | | | | | | | |
|  | 4.2 ± 1.6 | 4.8 ± 1.8 | 5.2 ± 1.5 | 4.9 ± 1.8 | 7.6 ± 1.8 | 6.0 ± 1.7 | 6.1 ± 1.4 | 8.9 ± 2.7 | 4.5 ± 1.8 |

Mean number of correct responses is indicated for the memory task ±SEM at training of the original task (Training), training of the interference task (Interference) and testing of the original task (Testing), as well as mean rating in the Stanford Sleepiness Scale (SSS) ± SEM and mean number of words subjects indicated of having heard during sleep in the Heard/not-heard task ±SEM for the three groups of experiments 2–4 in Study 2.
*Rc* complete reminder, *Ri* incomplete reminder, *NR* no reminder.

$F_{2,39} = 0.04$, $P = 0.96$, $\eta^2 = 0.002$; Table 2) and performance in the "heard/not-heard" task (one-way ANOVA, $F_{2,39} = 0.57$, $P = 0.57$, $\eta^2 = 0.028$; Table 2). With regard to time spent in different sleep stages, there was no significant difference for total time of sleep (one-way ANOVA, $F_{2,39} = 0.75$, $P = 0.48$, $\eta^2 = 0.037$) as well as for the percentage of wake (one-way ANOVA, $F_{2,39} = 0.16$, $P = 0.85$, $\eta^2 = 0.008$), stage 1 (one-way ANOVA, $F_{2,39} = 0.33$, $P = 0.72$,

$\eta^2 = 0.017$), stage 2 (one-way ANOVA, $F_{2,39} = 0.66$, $P = 0.52$, $\eta^2 = 0.033$), SWS (one-way ANOVA, $F_{2,39} = 0.20$, $P = 0.82$, $\eta^2 = 0.010$), and REM sleep (one-way ANOVA, $F_{2,39} = 0.91$, $P = 0.41$, $\eta^2 = 0.045$).

These results suggest that on the long-term, i.e., after an 8-h extended sleep interval, only incomplete reminders but not complete reminders induce memory stabilization.

**TABLE 3 Sleep parameters in Study 2.**

| | Exp. 2 (40 min) | | | Exp. 3 (8 h) | | | Exp. 4 (40 min-7h) | | |
|---|---|---|---|---|---|---|---|---|---|
| | Rc | Ri | NR | Rc | Ri | NR | Rc | Ri | NR |
| Wake | 3.7 ± 1.8 | 2.0 ± 0.9 | 3.0 ± 1.5 | 4.3 ± 1.0 | 5.6 ± 2.7 | 5.8 ± 1.9 | 2.1 ± 1.7 | 8.3 ± 3.7 | 3.2 ± 1.2 |
| Stage 1 | 10.6 ± 2.4 | 8.0 ± 1.4 | 8.5 ± 1.3 | 6.0 ± 0.6 | 5.1 ± 0.6 | 5.8 ± 1.0 | 4.1 ± 1.1 | 5.9 ± 1.3 | 4.3 ± 0.5 |
| Stage 2 | 36.7 ± 4.2 | 30.7 ± 2.9 | 37.6 ± 3.6 | 56.4 ± 1.4 | 54.8 ± 1.9 | 57.8 ± 2.1 | 39.9 ± 3.8 | 48.1 ± 4.4 | 53.3 ± 5.3 |
| SWS | 48.9 ± 5.9 | 53.3 ± 6.3 | 50.9 ± 4.9 | 15.0 ± 1.3 | 14.9 ± 1.7 | 13.7 ± 1.9 | 53.8 ± 4.7 | 36.7 ± 4.6 | 39.3 ± 5.6 |
| REM sleep | 0.0 ± 0.0 | 0.0 ± 0.0 | 0.0 ± 0.0 | 18.1 ± 1.1 | 19.4 ± 1.4 | 16.9 ± 1.4 | 0.0 ± 0.0 | 1.2 ± 1.2 | 0.0 ± 0.0 |
| TST (min) | 43.7 ± 3.4 | 47.8 ± 4.1 | 48.3 ± 2.6 | 458.0 ± 10.5 | 478.0 ± 10.2 | 460.1 ± 15.0 | 53.3 ± 3.6 | 85.5 ±12.7 | 64.8 ± 3.5 |

Percentage of total sleep time ±SEM is shown. Note that none of the participants in Exp. 2 (40 min) showed any signs of REM sleep and only one participant in the "incomplete reminder/40 min-7h" group of Exp. 4 spent 16 min in REM sleep.
*SWS* slow wave sleep (sum of stage 3 and stage 4), *REM sleep* rapid eye movement sleep, *TST* total sleep time, *Rc* complete reminder, *Ri* incomplete reminder, *NR* no reminder.

**Experiment 4: long-term memory stabilization with complete vs. incomplete reminders during SWS and additional wakefulness**. The procedures were basically identical to Experiment 2, except that participants remained awake for another 7 h after the sleep period with reactivation (Fig. 3a).

Learning performance did not differ between groups (one-way ANOVA, $F_{2,34} = 1.22$, $P = 0.31$, $\eta^2 = 0.067$; Table 2). However, significant differences emerged for memory change from the learning session to the testing session (Fig. 3d, one-way ANOVA, $F_{2,34} = 3.40$, $P = 0.045$, $\eta^2 = 0.167$). Here, we observed the same pattern of results as in Experiment 3, that is, the "incomplete reminder/40 min-7h" group performed significantly better than the "complete reminder/40 min-7h" group ($T_{22} = 2.31$, $P = 0.031$, $r = 0.44$) and tended to perform better than the "no reminder/40 min-7h" group ($T_{22} = 1.89$, $P = 0.072$, $r = 0.37$; "incomplete reminder/40 min-7h" $-4.0 ± 1.5$, "complete reminder/40 min-7h" $-8.2 ± 1.1$, "no reminder/40 min-7h" $-7.2 ± 0.9$). Performance of the "complete reminder/40 min-7h" group and the "no reminder/40 min-7h" group did not differ ($T_{24} = -0.71$, $P = 0.49$, $r = 0.14$).

There was no difference between groups in learning of the interference task (one-way ANOVA, $F_{2,34} = 1.25$, $P = 0.30$, $\eta^2 = 0.069$; Table 2), subjective sleepiness (one-way ANOVAs, for training: $F_{2,34} = 2.73$, $P = 0.08$, $\eta^2 = 0.138$; interference: $F_{2,34} = 2.95$; $P = 0.07$, $\eta^2 = 0.148$; testing: $F_{2,34} = 1.30$; $P = 0.29$, $\eta^2 = 0.071$; Table 2) and the "heard/not-heard" task (one-way ANOVA, $F_{2,34} = 1.20$, $P = 0.31$, $\eta^2 = 0.066$; Table 2). With regard to sleep stage distribution, there was a significant difference between groups in total time of sleep (one-way ANOVA, $F_{2,34} = 4.93$, $P = 0.013$, $\eta^2 = 0.225$), with the "complete reminder/40 min-7h" group showing less total sleep time than the "incomplete reminder/40 min-7h" ($T_{22} = -2.63$, $P = 0.015$, $r = 0.49$) and the "no reminder/40 min-7h" groups ($T_{24} = -2.28$, $P = 0.032$, $r = 0.42$). The percentage of SWS also differed (one-way ANOVA, $F_{2,34} = 3.39$, $P = 0.045$, $\eta^2 = 0.166$, with the "complete reminder/40 min-7h" group showing higher percentage of SWS than the "incomplete reminder/40 min-7h" group ($T_{22} = 2.57$, $P = 0.017$, $r = 0.48$) and the "no reminder/40 min-7h" group ($T_{24} = 1.99$, $P = 0.06$, $r = 0.38$). However, total time of sleep and percentage of SWS were not associated with memory change in any of the groups ($-0.24 < r < 0.4$, all $P > 0.22$). There was no significant difference between groups for percentage of wake (one-way ANOVA, $F_{2,34} = 1.99$, $P = 0.15$, $\eta^2 = 0.105$), stage 1 (one-way ANOVA, $F_{2,34} = 0.93$, $P = 0.41$, $\eta^2 = 0.052$), stage 2 (one-way ANOVA, $F_{2,34} = 2.30$, $P = 0.12$, $\eta^2 = 0.119$), and REM sleep (one-way ANOVA, $F_{2,34} = 1.20$, $P = 0.32$, $\eta^2 = 0.066$).

**Exploratory cross-experiment comparisons**. Although the three experiments of Study 2 cannot be directly compared because of different lengths of the retention interval (~2 h in Exp. 2 vs. ~9 h in Exp. 3 and 4) and different states at testing (rested in Exp. 2 and 3 vs. sleep deprived in Exp. 4), we conducted a cross-experiment analysis for exploratory purposes. It revealed significant differences for memory change between reminder types (two-way ANOVA, $F_{2,103} = 8.13$, $P = 0.001$, $\eta^2 = 0.136$) and experiments ($F_{2,103} = 5.66$ $P = 0.005$, $\eta^2 = 0.099$); yet, there was no significant interaction ($F_{4,103} = 1.66$, $P = 0.16$, $\eta^2 = 0.061$). Overall, the incomplete reminder groups showed better performance than the complete reminder groups as well as the no reminder groups (post-hoc LSD, $P = 0.001$ and $P < 0.001$, respectively). Furthermore, participants in Exp. 4 exhibited overall more forgetting than participants in Exp. 2 and Exp. 3 (post-hoc LSD, $P = 0.001$ and $P = 0.03$, respectively), which can be explained by participants being sleep deprived at testing in Exp. 4.

**Electrophysiological analysis of cued reactivation during sleep**. To test for differences in brain oscillation patterns upon reactivation with the different reminder types, time-frequency analyses were performed for each condition: Rc (complete reminder), Ri (incomplete reminder), and NR (no reminder). EEG data for the reactivation period were contrasted using equivalent time windows between the NR and Rc conditions (Fig. 4a), between the NR and Ri conditions (Fig. 4b), as well as between the Rc and Ri conditions (Fig. 4c). Following the presentation of the cue, both complete and incomplete reminder cues (i.e. words and syllables, respectively) evoked responses in the theta band (4–8 Hz) and the fast spindle band (12–16 Hz, all cluster-level $P < 0.001$) as compared to the no reminder condition. When both reminder conditions were directly compared, cue-evoked power increases in the slow spindle band (10–12 Hz) were stronger in the complete reminder compared to the incomplete reminder condition (Fig. 4c, $P = 0.025$), whereas power increases in the theta band (4–8 Hz) did not differ significantly ($P = 0.15$). Regarding responses evoked by the presentation of the sound (which was identical for both reminder conditions), significant power increases were detected in the fast spindle band in both reminder conditions (12–16 Hz, Rc vs. NR: Fig. 4a $P = 0.032$, Ri vs. NR: Fig. 4b; $P = 0.029$); yet despite descriptive increases in theta power in response to the sound, there were no significant clusters spanning that frequency range (Rc vs. NR: $P = 0.40$, Ri vs. NR: $P = 0.95$). Interindividual differences and variability in the sound recordings may have introduced variance obscuring sound-induced power changes, which have typically been observed in previous studies using sounds as reactivation cues[46]. However, a genuinely attenuated theta response would also be in line with previous research, suggesting that additional sounds immediately following a reminder cue can disrupt processing of that initial reminder[47,48]. Long-lasting auditory stimulation like the sounds used in the present study may yield a similar effect.

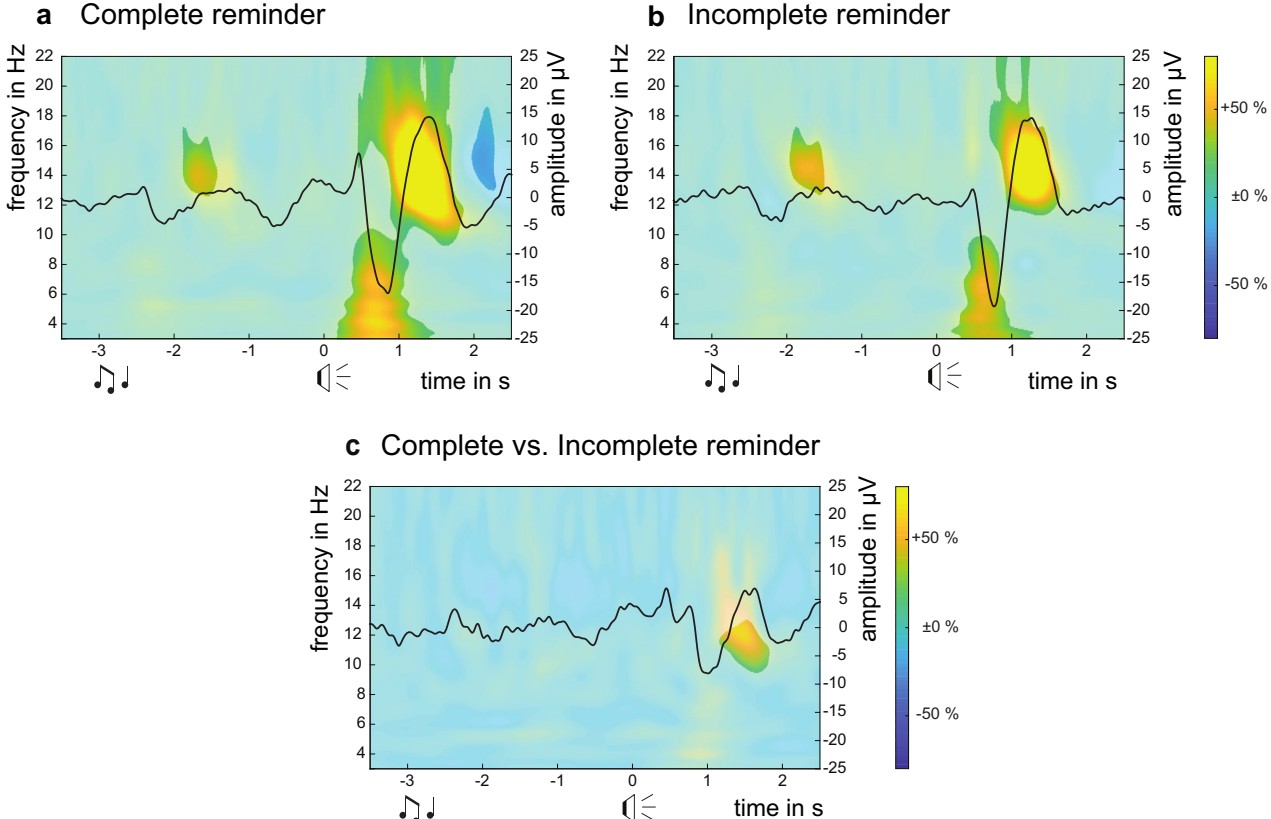

**Fig. 4 Reminder cues during SWS evoke responses in the theta and spindle band.** EEG power response over central sites aligned to cue onset ($t = 0$ s, representing onset of the entire word for the complete reminder and the first syllable for the incomplete reminder, respectively). Note that reminder conditions are identical during sound presentation and only differ from the start of cue presentation. Grand averages from all experiments of Study 2 are shown for **a** the complete reminder conditions and **b** the incomplete reminder conditions. Colors show power changes relative to pre-stimulation baseline between −4 and −3 s. Black waveform shows evoked time-domain response. Data are masked by cluster permutation statistics contrasting stimulation time windows to comparable time windows without stimulation in the no reminder conditions. Cluster-level statistics (sample-level threshold of $P = 0.01$) for **a**: sound-evoked spindle cluster $P = 0.032$, cue-evoked spindle and theta clusters $P < 0.001$; for **b**: sound-evoked spindle cluster $P = 0.029$, cue-evoked spindle and theta clusters $P < 0.001$. **c** Comparison between data from both conditions (Incomplete reminder subtracted from and statistically compared to Complete reminder). Cluster-level statistics (using a sample-level threshold of $P = 0.01$) for cue-evoked spindle cluster: $P = 0.025$. Musical notes represent onset of the sound, loudspeakers represent onset of the cue (word/syllable).

**Correlation analyses.** To test whether the observed behavioral effects in memory performance are associated with changes in sleep measures, we performed correlation analyses between memory change and time spent in single sleep stages as well as power in specific frequency bands during the entire sleep period. In the "no reminder/8 h" group, higher scores in memory change were associated with higher percentage of SWS (Fig. 5a, $r = 0.66$, $P = 0.008$), as well as with higher slow oscillation power (0.5–1 Hz; Fig. 5b, $r = 0.55$, $P = 0.03$), higher delta power (1–4 Hz; Fig. 5c, $r = 0.56$, $P = 0.0029$), and higher spindle power (9–15 Hz; Fig. 5d, $r = 0.64$, $P = 0.01$). In the "no reminder-40 min" group, memory change was only associated with higher spindle power (9–15 Hz; Fig. 5e, $r = 0.64$, $P = 0.035$). There were no other significant correlations. We further analyzed correlations between memory change and the observed spindle and theta power increases upon reactivation with the different reminder cues (as reported in Fig. 4). This analysis revealed no significant correlations, except for an association of memory change with the cue-evoked theta power increase in the "complete reminder-40 min/7 h" group ($r = −0.64$, $P = 0.047$). However, it should be noted that none of the correlations were corrected for multiple comparisons and should therefore be interpreted with caution.

## Discussion

We here show that memory stabilization by cueing reactivation during sleep depends on the type of reminder cue as well as the time interval between reactivation and testing. Incomplete reminders and complete reminders were equally effective in stabilizing memories at short term, when testing took place immediately after reactivation. However, when testing was delayed by 7 h, only the incomplete reminder but not the complete reminder resulted in memory stabilization, with this effect being independent of whether the delay interval was filled with additional sleep or wakefulness.

First, we established that only incomplete reminders but not complete reminders are capable of inducing memory labilization/reconsolidation in the wake state. This finding is consistent with previous evidence from the reconsolidation literature[8], suggesting that a prediction error, i.e., the mismatch between what is predicted according to previous stimuli contingencies and what is actually encountered during reminder presentation, determines whether a memory trace becomes labile or not[9–11,13–15]. In the present study, the incomplete reminder was formed of the sound plus the first syllable of the associated word, with the entire word never being presented. This might have violated the subjects' predictions and thereby generated a mismatch.

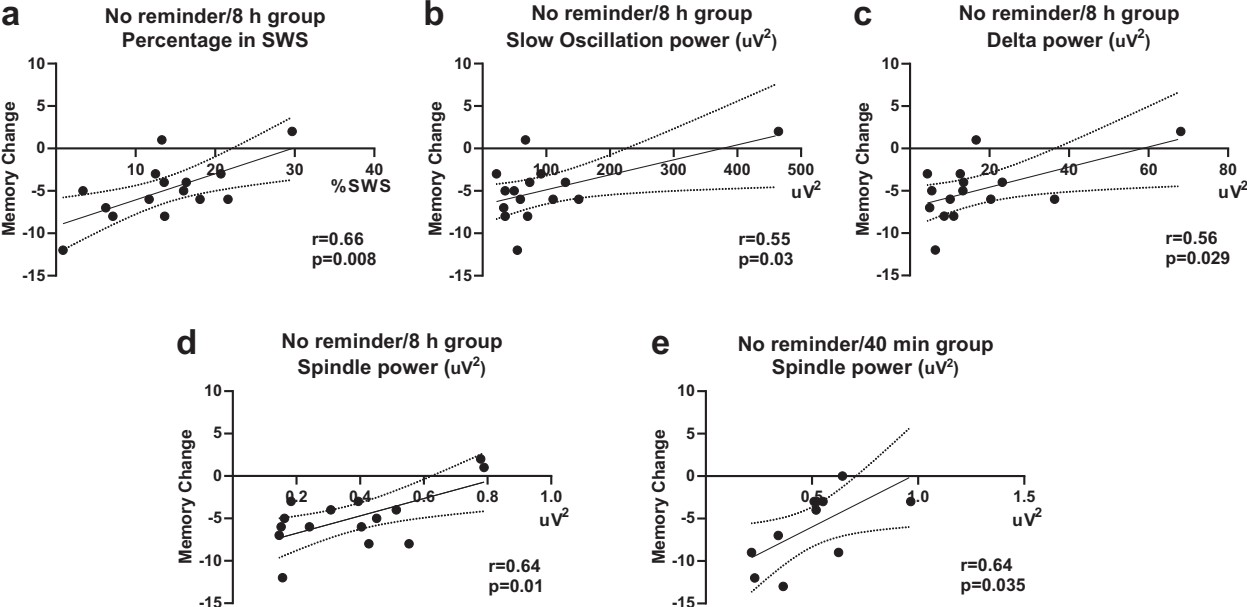

**Fig. 5 Sleep and memory performance.** Significant correlations in the "no reminder/8 h" group between memory change and **a** percentage of SWS, **b** slow oscillation power in NREM, **c** delta power in NREM, and **d** spindle power in NREM. **e** Significant correlation in the "no reminder/40 min" group between memory change and spindle power in NREM.

As for cueing memory reactivation during sleep, we hypothesized that only incomplete but not complete reminders would stabilize the reactivated memories. Our results only partly support this hypothesis: incomplete reminders were indeed more effective in stabilizing memories for the long-term; yet both reminders resulted in memory stabilization when tested at short-term. The finding that incomplete reminders stabilize memories already at short-term is in contrast with evidence from reconsolidation studies in the wake state. For reconsolidation, the strengthening/impairing effects of incomplete reminders are typically only observed at long-term when the memories are already re-stabilized, but not at short-term when they are still labile[2,6,49]. Our finding of an immediate stabilization upon cueing during sleep suggests that sleep speeds up memory (re-)stabilization, such that the beneficial effects of incomplete reminders are already observed after shorter time intervals. In accordance, we have recently shown that sleep accelerates memory restabilization, shortening the time window during which a reactivated memory is susceptible to interference[50]. In the present study, the reactivated memories might have been re-stabilized already within the ~40-min sleep period, producing the short-term stabilization observed right after this sleep period.

The finding that complete reminders likewise stabilized memories at short-term, also contrasts with the literature on wake reconsolidation. In reconsolidation paradigms, the impairing effects as well as the strengthening effects are typically only observed for incomplete reminders but not for complete reminders[8,13,14,49]. In the present study, the stabilizing effect of complete reminders was only evident at short-term and disappeared over time, whereas incomplete reminders resulted in both short-term and long-term stabilization. It is conceivable that both reminders facilitate early consolidation processes involving short-lasting functional and/or structural synaptic changes that are independent of protein synthesis[51]. These early consolidation processes take place within ~30 min and induce a fast increase in synaptic strength[51,52], possibly resulting in the stabilization of the memory traces against interference at short-term. However, these early changes are transient and decay after ~90 min[53]. Additional

processes of late consolidation may be necessary to stabilize the memories for the long-term, presumably depending on protein synthesis and the availability of plasticity-related proteins[54]. For its induction, it may be essential for the memories to be tagged during the initial cueing period[51,55], which may only be achieved by incomplete reminders.

One explanation for why only incomplete reminders achieve this tagging is related to the mismatch hypothesis proposing that incomplete reminders induce a prediction error that then triggers the labilization and subsequent updating or strengthening of the memory trace[7,14,56]. We propose that prediction errors can also be detected during sleep upon encountering incomplete reminder cues, tagging the respective memories for late consolidation. While the P300 and the intermediate mismatch peak of the MMN were found to be abolished during sleep, the early and late parts of the MMN are preserved during sleep[45]. Although most studies focused on mismatch detection during light sleep stages[45,57], some studies observed signs of mismatch detection also during SWS[58–60], suggesting that at least a rudimentary form of mismatch detection may still be in place during SWS. Unfortunately, the present experimental design did not allow for a direct analysis of mismatch detection during sleep with event-related potentials analyses because of a low number of reactivation trials, varying lengths between cues (syllables/words for incomplete/complete reminders) and the number of rejected trials after data preprocessing. However, we showed that both the incomplete and complete reminder cues evoked consistent increases in theta band power, followed by responses in the spindle band. These results are in line with prior reports from auditory cueing studies[47,48,61,62]. Interestingly, there were stronger spindle-band responses to the complete reminder when compared to the incomplete reminder, which indicates differences in neuronal processing of complete and incomplete reminders during sleep. Complete reminder cues are semantically richer and may trigger a broader set of neuronal processes that entail the full (or larger parts of the) memory representation. Incomplete cues, on the other hand, resulted in weaker responses, indicating that these cues only activate a subset of the learned information, thereby

potentially inducing a prediction error. However, it is also known that sleep spindles play a role in preserving sleep[63]. Since complete reminders were of slightly longer duration, these reminders might have a higher tendency of inducing arousals and thus elicit stronger spindle responses to prevent subjects from waking up. Thus, the difference in responses between incomplete and complete reminder cues should be interpreted with caution.

Although we did not find any consistent associations between the spindle and theta responses upon reminder presentation and memory performance, we did observe associations between memory performance and other sleep measures. Better memory performance was associated with higher amounts of SWS as well as with higher power in the slow oscillation, delta and spindle band in the "no reminder/8 h" group. In the "no reminder/40 min" group, better memory performance was associated with power in the spindle band. Interestingly, correlations with memory performance were only observed in the groups that received no reactivation. Indeed, it is well known that the time spent in SWS as well as the power of slow oscillations, delta and spindles is related to sleep-dependent memory improvement[64]. It was also found that declarative memory consolidation particularly depends on the duration of SWS and that external reactivation cues can accelerate the consolidation process[65]. It could be speculated that without externally triggered reactivation, the amount of SWS as well as associated slow oscillations, delta and spindle activity, makes a difference for memory consolidation because of more or less spontaneous reactivation events. Externally triggered reactivation may speed up this process and facilitate stronger "artificial" reactivations such that the improvement of performance does not depend on spontaneous reactivations so much anymore. This idea is in keeping with Lewis and Bendor's model[66]. They propose that external sensory cues are processed in the neocortex during sleep, which effectively tricks the hippocampus by evoking what looks like a spontaneous reactivation of a memory in the cortex, in turn influencing hippocampal replay[66].

In accordance with our hypothesis, during sleep, only incomplete but not complete reminders resulted in long-term memory stabilization. Interestingly, these long-term stabilization processes seem to be initiated by reactivation during the first ~40 minutes of sleep and then continue to develop independent of whether the ensuing 7-h period is spent awake or asleep. It could be speculated that late consolidation processes triggered by mismatch-inducing incomplete reminders during sleep depend on molecular and cellular processes that can be effectively established during both sleep and wakefulness. One candidate mechanism are plasticity-related proteins, such as the immediate early genes *Arc* and *Zif-268*[67,68], which are selectively upregulated during post-learning REM sleep as well as during wakefulness, while they are downregulated during SWS[69–71]. These processes may be complemented by other factors, such as high levels of acetylcholine during both REM sleep and wakefulness, as opposed to low cholinergic tone during SWS[64]. These ideas are reminiscent of earlier two-stage models, suggesting that successful memory consolidation depends on the cyclic succession of SWS and REM sleep[72,73], with SWS supporting early consolidation and REM sleep late consolidation. Our data extend this view in proposing that late periods of wakefulness may be similarly effective than REM sleep in supporting memory stabilization for the long-term. However, assuming that the memory benefit in the incomplete reminder condition is established already after ~40 min of sleep and remains unchanged thereafter, the difference between reminder types could be alternatively explained by a memory decay for the complete reminder condition during the extended retention interval. These two alternative explanations cannot be discerned with the present experimental design and should be

subject to further investigation. Generally, the comparison between experiments should be interpreted with caution because there are several differences between experiments including circadian effects, tiredness, sleep length, and temporal proximity between training, reactivation, and interference/testing, all of which are known to affect memory performance. Particularly relevant seems to be the difference in temporal proximity between reactivation and interference/testing (~30 min in Exp. 2 vs. ~8 h in Exp. 3 and 4), differences in total sleep time (~47 min in Exp. 2, ~465 min in Exp. 3, and ~67 min in Exp. 4) as well as different states of tiredness at testing (rested in Exp. 2 and 3 vs. sleep deprived in Exp. 4), adding further limitations to the study.

Two recent studies investigated the impact of auditory feedback presented during NREM sleep. Schreiner and colleagues[47] applied either single cues (i.e. Dutch words) or cues with feedback (i.e. Dutch-German vocabulary) and found that only the single cue condition, i.e. cueing with an incomplete reminder, produced a memory improvement, whereas presenting feedback immediately after the cue blocked this improvement. In a similar study by Farthouat et al.[48], presenting cues with feedback (i.e. word pairs) did not result in any memory improvement (yet, this study did not include a single cue condition). These findings are similar to the results of our long-term experiments (Exp. 3 and 4), with only the incomplete but not the complete reminders producing memory benefits. Although in the Schreiner et al. and the Farthouat et al. studies, participants slept for only 3 h and 90 min, respectively, this is still substantially longer than the ~40-min sleep period in our short-term experiment (Exp. 2). A longer sleep period of 90 min to 3 h, also including a fair amount of REM sleep, thus, appears to be sufficient for long-term (i.e. late) consolidation effects to become evident similar to our findings of the long-term experiments. Alternatively, it could be argued that in the present study, it is not the incomplete aspect of the reminder, but rather the interference between different parts of the complete reminder, that best explains the lack of a cueing effect for complete reminders. This interpretation is based on an additional finding by Schreiner and colleagues[47], showing that the presentation of a tone after the single cue likewise inhibits consolidation. We suggest that it may be a combination of both processes that determines the reactivation effect, i.e. the incomplete aspect of the cue and a silent refractory period after the incomplete cue. The cueing effect may not be observed when either a complete cue is presented, because complete cues do not induce a mismatch, or an incomplete cue is presented with a subsequent sound, because then the sound disrupts the processing of the incomplete cue. Future studies should vary the type of reminder with or without an interruption of the refractory period and examine the exact time frame and the contribution of different sleep stages for memory reactivation during sleep.

## Methods

**Subjects**. Altogether 214 participants were enrolled in the study and provided written informed consent prior to participation. All experiments were approved by the ethics committee of the Medical Faculty of the University of Tübingen. The methods were carried out in accordance with the relevant guidelines and regulations and participants received a financial compensation of 30 Euro for participating in Study 1 and 80 Euro for Study 2. Participants were recruited by advertisement at the University of Tübingen. None of the participants reported being sick during the experiment, to have any mood disorder or any anxiety disorder, any history of neurological diseases, did not take any medication at the time of the experiments, did not suffer from any sleep disorders, had not been on night shift and did not have any major sleep disturbances during the 6 weeks prior to the experiments. Female participants had a regular menstrual cycle according to self-reports and did not take oral contraceptives. The experiment was scheduled to take place in the luteal phase of the menstrual cycle for all women (15–28th day), based on previous evidence that sleep effects for memory are more readily observed in this phase[74].

Sample size was determined based on power analyses with expected effect sizes estimated from previous experiments from our groups[8,15,38]. Participants were

randomly assigned to each group, however, the experimenter was aware of the group assignment because he/she had to administer the correct reactivation protocol during wakefulness or sleep.

A total of 55 subjects participated in Study 1, with 7 subjects being excluded from data analysis because they did not reach the learning criterion during the training session (3 subjects), because of technical problems (2 subjects), because they did not understand the instructions of the experiment (1 subject), or because they did not complete all experimental sessions (1 subject); leaving 48 subjects for the final analysis (mean age: 23.5 ± 0.5; 24 females).

Overall 159 subjects participated in Study 2. Of these, 47 subjects participated in Experiment 2, with 14 subjects being excluded from the analysis because of the following reasons: technical problems (2 subject), difficulties to fall asleep (4 subjects), waking up during reactivation (6 subjects), not completing all experimental sessions (1 subject), or snoring-induced repetitive EEG arousals (1 subject). Data from the remaining 33 subjects were included in the final analysis (mean age: 23.6 ± 0.6; 21 females). A total of 59 subjects participated in Experiment 3, with 17 subjects being excluded from data analysis because of technical problems (4 subjects) or because they did not reach the learning criterion (1 subject), had difficulties to fall asleep (4 subjects), woke up during reactivation (5 subjects), or did not complete all experimental sessions (3 subject); leaving 42 subjects for the final analysis (mean age: 23.3 ± 0.4; 32 females). A total of 53 subjects participated in Experiment 4, with 16 subjects being excluded from data analysis because of technical problems (2 subjects) or because they did not complete all experimental sessions (5 subject), had difficulties to fall asleep (2 subjects), woke up during reactivation (5 subjects) or fell asleep during the 7-h-wake interval (2 subjects); leaving 37 subjects for the final analysis (mean age: 23.1 ± 0.5; 22 females).

**Experimental procedures**. All subjects took part in a training session (during which the original task was learned), a reactivation session (during which different types of reminders were presented that were or were not followed by an interference task), and a testing session (during which the recall of the original task was assessed; see Fig. 1).

**Study 1 – reconsolidation during wakefulness**. This study was performed during the day-time (8:00–16:00 h) on 3 consecutive days (Fig. 2a). Each participant arrived at the laboratory at the same time of day on all of the three days. On Day 1, participants signed the informed consent, filled out a personal data questionnaire and the Stanford Sleepiness Scale. Afterwards they performed the training session that was introduced by a demo-trial in order for subjects to understand the goal of the task. After the training session, subjects left the laboratory and returned on the next day. On Day 2, after completing the Stanford Sleepiness Scale again, participants took part in the reactivation session, during which they received a reminder of the learning task (complete reminder or incomplete reminder, see below). The reactivation session was or was not followed by an interference task to impair memory restabilization. The "complete reminder/no interference" group ($n = 11$, age: 23.4 ± 0.6, 5 females), received the complete reminder without the interference task; the "complete reminder/interference" group ($n = 12$, age: 22.8 ± 0.6, 5 females) received the complete reminder followed by the interference task; the "incomplete reminder/no interference" group received the incomplete reminder without the interference task ($n = 13$, age: 23.8 ± 0.9, 7 females); the "incomplete reminder/interference group" ($n = 12$, age: 24.1 ± 1.0, 7 females) received the incomplete reminder followed by the interference task. On Day 3, subjects completed the Stanford Sleepiness Scale again and then participated in the testing session, during which their memory of the originally learned task from Day 1 was assessed. Finally, subjects filled out a questionnaire on their sleep behavior and activities during the experimental days.

**Study 2 – cued memory reactivation during sleep**
*Experiment 2*. Subjects first spent an adaptation night in the sleep lab including placement of electrodes before the experimental night. The adaptation night and the experimental night were separated by at least 48 h. For the experimental night, subjects participated in one of three groups ($n = 11$ each), the "complete reminder" group (age: 24.5 ± 1.0, 7 females), the "incomplete reminder" group (age: 23.9 ± 0.9, 6 females), or the "no reminder" group (age: 22.5 ± 0.7, 8 females; Fig. 3a). All subjects arrived at the laboratory at 21:00 h and were prepared for polysomnographic recordings. At 22:15 h they filled out the Stanford Sleepiness Scale and the training session started at 22:30 h. At ~23:00 h participants were allowed to sleep in a quiet, darkened room. Participants were presented with white noise (43 dB) via in-ear headphones from the time they went to bed until they were awakened. The in-ear headphones were attached to the ear with a hypoallergenic tape and it was ensured that the headphones were still in place upon waking up. The experimenter monitored the sleep EEG online and started the reactivation after 10 min of stable SWS. Participants were awakened after ~40 min of sleep (counting from the first sleep spindle or K-Complex online detected) and electrodes were removed. About 30 min after being awakened, all participants learned the interference task and filled out the Stanford Sleepiness Scale again. Thirty minutes after the interference task was finished, participants completed the testing session, in which their memory of the original task was assessed. Please note that the interference task in Study 2 was not used to impair memory stabilization as in Study 1, but rather to enhance the

contrast between reminder conditions in the subsequent test. Finally, subjects filled out the Stanford Sleepiness Scale one last time and completed a final questionnaire as well as the "heard/not-heard" task to test whether they had heard any of the stimuli during sleep. Participants then continued sleeping in the lab until the next morning.

*Experiment 3*. In this experiment, the procedures were basically identical to Experiment 2, except that participants were left sleeping for additional 7 h during the experimental night after the first ~40 min sleep period during which the reactivation occurred. Importantly, no further reminders were presented during the additional 7 h sleep interval. In all, 30 min after being awakened, the participants received the interference task and 30 min later they were tested on the original task. Like in Experiment 2, subjects participated in one of three groups, the "complete reminder" group ($n = 14$, age: 23.5 ± 0.6, 11 females), the "incomplete reminder" group ($n = 13$, age: 21.8 ± 0.8, 9 females), or the "no reminder" group ($n = 15$, age: 24.2 ± 0.7, 12 females; Fig. 3a).

*Experiment 4*. In this experiment, the procedures were basically identical to Experiment 3, except that participants were awakened after the 40-min sleep reactivation period, electrodes were removed and they remained awake for another 7 h, playing board games and reading. Like in Experiment 3, no further reminder cues were presented during the additional wake period. After that period, the interference task and testing session took place like in Experiments 2 and 3. Again, subjects participated in one of three groups, the "complete reminder" group ($n = 13$, age: 22.3 ± 0.8, 8 females), the "incomplete reminder" group ($n = 11$, age: 23.6 ± 1.0, 9 females), or the "no reminder" group ($n = 13$, age: 23.5 ± 0.8, 5 females; Fig. 3a).

**Memory task**. The task required participants to associate 30 specific sounds with 30 German words. The sounds and the words were semantically related (for example, sound of a storm associated with the word LAWINE [avalanche]), given that cueing effects have been suggested to be particularly strong for learning associations that are based on prior knowledge[75]. Each sound had a duration between 2855 and 2940 ms (on average 2900 ms) and was repeated in a loop until the specified presentation time was reached. All words had three syllables and were pre-recorded with a female voice (Fig. 1).

**Training session**. All of the 30 sound-word associations were presented one after the other. Each trial started with the presentation of the sound for on average of 2900 ms (via headphones). The sound then continued accompanied by the associated word written on the screen and spoken aloud once via headphones (1500 ms; Fig. 1). After 4000 ms break, the next sound and word appeared. Once all of the 30 sound-word pairs had been presented, participants' learning success was assessed in a cued recall. Each sound appeared an average of 2900 ms and continued accompanied by the first syllable of the associated word presented aloud once via headphones (1500 ms). After that, the sound continued and the image of a microphone appeared on the screen indicating that subjects had to answer by saying the associated word aloud. After 5000 ms, the sound plus the correct word was presented written on the screen and aloud via headphones to provide correct feedback (1500 ms). The same procedure was repeated after a break of 4000 ms for all of the 30 sound-word associations. The training session took ~15 min. Subjects that did not reach 40% correct responses (12 correct answers) were excluded from the analysis.

**Reactivation session**. During the reactivation session, two different types of reminders were presented, the complete reminder ($R_c$) or the incomplete reminder ($R_i$; Fig. 1). In Study 1, subjects were instructed that this session would be similar to the cued recall session and that they should say the associated word aloud after the presentation of the sound plus the first syllable once the microphone appeared on the screen. However, the microphone never appeared, thus subjects never actually gave a response. The complete reminder condition included the presentation of the sound (on average 2900 ms) and the sound then continued accompanied by the entire word presented aloud via headphones for 1500 ms. After that, a message of interruption ("trial interrupted") appeared on the screen (4000 ms) followed by another message indicating that the experiment will continue ("trial continues") with the next trial (3000 ms). Then the next sound-word association was presented in the same fashion until all of the 30 associations were shown once. Importantly, subjects were instructed that a number of different messages could appear on the screen and that they should read and pay attention to the messages and should not open the door and talk to the experimenter until the message "end of the experiment" was shown. The incomplete reminder condition was identical to the complete reminder condition except that instead of the entire word, only the first syllable of each word was presented followed by the interruption sign. The reactivation session took ~6 min. The order of sound-word associations was pseudo-randomized across all training and reactivation trials, but was the same for all subjects. It is important to highlight that the reactivation session differed from the cued recall procedure during the training session in two regards: (1) participants expected to respond as in the cued recall procedure but the microphone indicating the opportunity to respond never appeared on the screen and (2) each trial was interrupted without allowing the participants to respond, a procedure known to

induce a prediction error, which triggers memory labilization/reconsolidation[8,15]. After the reactivation session, almost all participants reported to the experimenter that the microphone did not appear on the screen and that they could never respond, some of them even thought that there was a technical problem, suggesting that subjects actually expected to be required to respond until the end of the reactivation session.

In Study 2, for the sleep reactivation protocol, the same reminder conditions were used as in Study 1. Each sound was presented for an average of 2900 ms (45 dB) and the sound then continued accompanied by the cue (i.e. the entire word for the complete reminder condition or the first syllable for the incomplete reminder condition, 45 dB) presented aloud once via earphones for 1500 ms. After a seven-second interval, the next sound-cue was presented until all of the 30 associations were presented once. In all experiments, reminders were presented only in SWS during the first ~40 min of sleep. The ~40-min sleep duration was counted from the first spindle or K-complex detected by the experimenter online. For groups that received the reactivation during SWS, the experimenter had to wait 10 min after the subjects reached stable SWS before starting the reactivation. Thus, in some cases the experimenter had to wait longer until the participant reached stable SWS (amounting to longer sleep duration in some cases). Moreover, the reactivation was paused whenever signs of arousals or changes in sleep stage were detected, and resumed upon stable SWS was detected again, additionally adding some variance in individual sleep time. Furthermore, if participants did not reach stable SWS again after 90 min, the reactivation did not continue to prevent reactivation in the second sleep cycle, and the participant was excluded from the experiment. Each cue was presented once during reactivation, that is, all participants received 30 reactivation trials. The entire reactivation procedure took about 5 min and 45 s. Participants were awakened after the reactivation procedure (i.e. 30 reactivation trials) was completed and they had slept for at least 40 min (Exp. 2 and 4) or 8 h (Exp. 3). After awakening, electrodes were removed. Subjects did not receive any instruction before they went to bed but they were told that some of the learned sounds and words could be played to them during sleep via earphones.

**Interference task**. The interference task consisted of the same sounds from the training session but those sounds were now associated with other words. These words were also semantically related to the sounds and likewise consisted of three syllables, but with a different first syllable than the words of the original task. The procedure was the same as for the training session.

**Testing session**. For each sound-word association, the sound was played for 5500 ms. In all, 500 ms into the sound, the image of a microphone appeared upon which subjects had 5000 ms to say the associated word of the first-learned list aloud (Fig. 1). After a break of 4000 ms, the procedure continued until all of the 30 associations were presented.

**Sleep data**. Sleep was recorded by standard polysomnography including electroencephalographic (EEG), electromyographic (EMG), and electrooculographic (EOG) recordings with BrainAmp amplifiers (Brain Products, Munich, Germany). EEG was recorded from six scalp electrodes (F3, F4, C3, C4, P3, and P4 according to the International 10–20 System) and two electrodes on the left and right mastoids serving as combined reference. Data were recorded at a sampling rate of 200 Hz and bandpass-filtered between 0.16 and 35 Hz. In addition to the online identification of sleep stages, polysomnographic recordings were scored offline according to standard criteria as wake, sleep stages 1–4 and rapid eye movement (REM) sleep, with sleep stages 3 and 4 defining SWS[76].

**Reactivation EEG data processing**. All experiments in Study 2 diverged only after the reactivation phase. Therefore, electrophysiological data for the reactivation period of the three experiments were pooled for each condition, resulting in three datasets: complete reminder (Rc), incomplete reminder (Ri), and no reminder (NR). The onset of each stimulus (sounds and cues) was automatically marked in the EEG recording during reactivation. For the groups that did not receive any reactivation (NR groups), markers were assigned post-hoc according to the same criteria (i.e. starting after 10 min of stable SWS during the first ~40 min of sleep, without overlapping the segments of reactivation). EEG data of each reactivation event was cut into 12-s trials ($-7–5$ s, with $t = 0$ s referring to cue onset). Because the reminder conditions are identical during the sound presentation interval and only differ upon presentation of the cue (syllable/word), the analyses were time-locked to cue onset. Trials containing artefacts in any of the channels were removed using automatic and visual rejection. After data preprocessing, EEG data from 108 subjects remained for further analyses, overall 16 reactivation trials were excluded for Rc, 10 trials for Ri, and 12 trials for NR. Time-frequency analysis was performed (10-cycle Morlet wavelets) to calculate the power for the average of the electrodes C3 and C4 relative to a baseline of one second right before sound onset ($-4$ and $-3$ s). This processing resulted in the relative power of each frequency at each time point for Rc, Ri, and NR independently. All reactivations were aligned to cue onset ($t = 0$ s). Significant responses upon the sound and cue presentation in the Rc and Ri conditions were compared to the NR condition as well as between both reminder conditions. Responses upon the sound presentation are called "sound-evoked" responses and responses upon the cue presentation are called "cue-evoked" responses. Different

significant clusters were detected, according to the classical band frequencies for slow spindles (10–12 Hz), fast spindles (12–16 Hz), and theta (4–8 Hz).

**Correlations with sleep stages and power**. We further performed Pearson correlations between percentage in different sleep stages and memory change for all the groups of Study 2. Moreover, we performed Pearson correlations between memory change and power in the frequency bands of interest (slow oscillation: 0.5–1 Hz; delta: 1–4 Hz; theta: 4–8 Hz; and spindles: 9–15 Hz) during NREM sleep for all groups of Study 2.

For calculating individual average power spectra, the data were analyzed using SpiSOP (https://www.spisop.org, RRID:SCR_015673) that is based on code of FieldTrip[77] (http://fieldtriptoolbox.org, RRID:SCR_004849) in MATLAB 2013b (Mathworks, Natick, USA). Artifact-free NREM epochs were divided into consecutive 10 s blocks that overlapped 5 s in time. Each block was tapered by a single Hanning window before applying Fast Fourier Transformation that resulted in block power spectra with a frequency resolution of 0.1 Hz. Power spectra were then averaged across all blocks (Welch's method). Mean power over central electrodes was determined for the frequency bands of interest, slow oscillations (0.5–1 Hz); delta (1–4 Hz), theta (4–8 Hz), and spindle range (9–15 Hz).

**Control measures**

*Stanford Sleepiness Scale*. Subjects rated their subjective sleepiness on a scale ranging from 1 ("feeling active, vital, alert or wide awake") to 7 ("no longer fighting sleep, sleep onset soon; having dream-like thoughts").

*Heard/not-heard task*. In Study 2, after the testing session, participants were asked whether they had heard any sounds or words while they were sleeping. Additionally, all sounds plus words (in the complete reminder condition) or all sounds plus first syllables (in the incomplete reminder condition) were presented again and subjects had to indicate if they had received those stimuli while they were sleeping.

**Statistics and reproducibility**. For all experiments, answers in the memory task were counted as correct only if they completely matched the learned words. Memory change, i.e., the number of correct responses at testing minus the number of correct responses at training, was calculated as the performance measure. Study 1 included the "complete reminder/no interference" group ($n = 11$), the "complete reminder/interference" group ($n = 12$), the "incomplete reminder/no interference" group ($n = 13$), and the "incomplete reminder/interference" group ($n = 12$). Data were analyzed with ANOVA, with the between-subjects factors "reminder type" (complete reminder vs. incomplete reminder) and "interference" (interference vs. no interference). In case of a significant interaction, simple effect analyses were performed. In Study 2, all experiments included three groups. Experiment 2: "complete reminder/40 min" group ($n = 11$), "incomplete reminder/40 min" group ($n = 11$), "no reminder/40 min" group ($n = 11$), Experiment 3: "complete reminder/8 h" group ($n = 14$), "incomplete reminder/8 h" group ($n = 13$), "no reminder/ 8 h" group ($n = 15$), Experiment 4: "complete reminder/40 min-7h" group ($n = 13$), "incomplete reminder/40 min-7h" group ($n = 11$), and "no reminder/40 min-7h" group ($n = 13$). The experiments were analyzed separately with ANOVAs, with the between-subjects factor "reminder type" (complete reminder, incomplete reminder, and no reminder), followed by post-hoc t-tests. Control variables, i.e., subjective sleepiness, sleep data, and the Heard/not-heard task (i.e., mean number of correctly identified associations) were analyzed with one-way ANOVA. For additional exploratory cross-experiment analysis, a two-way ANOVA with the between-subjects factors "reminder type" (complete reminder, incomplete reminder, and no reminder) and "experiment" (experiment 2, 3, and 4) was conducted, followed by post-hoc tests. Analyses were performed with SPSS (version 22.0). Alpha was set to 0.05.

In order to examine if the time spent in the different sleep stages and power of the frequencies of interest were related to memory change, we performed bilateral Pearson correlations. The correlations were not corrected for multiple comparisons.

To assess the statistical differences between the EEG response to the reactivation in each condition, the time-frequency values previously calculated (see Reactivation EEG processing) were compared using sample-level two-tailed independent-samples t-tests followed by a non-parametric cluster-permutation procedure to correct for multiple comparisons[78] (5000 permutations), as implemented in the open-source toolbox FieldTrip[77]. All EEG data processing was done using Matlab 2017a (MathWorks Inc, Natick, Massachusetts, USA).

We did not explicitly replicate our findings. However, Study 1 can be considered an indirect replication of previous findings from our group[8,15].

**Reporting summary**. Further information on research design is available in the Nature Research Reporting Summary linked to this article.

## Data availability

Raw data of study 1 and 2 are included in Supplementary Data 1 and 2, respectively. The authors declare that data supporting the findings of this study are available from the corresponding author upon request.

## Code availability

The authors declare that the code for power analyses is open source (SpiSOP, https://www.spisop.org, RRID:SCR_015673) and together with the code for the time-frequency analyses of the reactivation trials is available at https://osf.io/5jvun (ref. [79]).

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

## Acknowledgements

The authors would like to thank Katharina Zinke for recording the audio files used in the paradigm. This work was funded by a grant from the Deutsche Forschungsgemeinschaft (DFG, TR-SFB 654) to J.B. and S.D., a collaborative grant from the DFG and CONICET/MINCYT to S.D. (DI 1866/2-1) and C.F. (Resolución D. No 4427) and a postdoctoral research fellowship from the Deutscher Akademischer Austauschdienst (DAAD) to C.F. The experiments were performed while C.F. was affiliated with the Institute of Medical Psychology and Behavioral Neurobiology, University of Tübingen, Tübingen, Germany.

## Author contributions

C.F., S.D., and J.B. conceived the idea, C.F., M.R., and J.C. collected the data, C.F., J.G.K., J.C., F.D.W., and S.D. analyzed the data, all authors wrote and reviewed the manuscript.

## Funding

## Competing interests

The authors declare no competing interests.
