## [Peer Review File · Communications Biology]

Reviewers' comments:

Reviewer #1 (Remarks to the Author):

Forcato and colleagues explore the parallels between memory reconsolidation during wake and consolidation during NREM sleep by running multiple experiments exploring the benefits of complete and incomplete reminders to memory during wake and sleep. Drawing from the literature on targeted memory reactivation, they conclude that incomplete - but not complete - reminders during sleep lead to prolonged benefits to memory. The manuscript is well written and explores a previously unaddressed question using appropriate methods and analyses. There are some minor issues regarding the theoretical background and interpretation that need to be addressed before it should be accepted for publication in this journal.

Major issues:

A. The main hypothesis in this manuscript is that the cognitive mechanism involved in reconsolidation during wake may similarly affect consolidation during sleep. In reconsolidation paradigms, partial cuing during wake labilizes memories and allows updating. However, as the authors themselves show, reconsolidation using partial cues is not necessary for memory strengthening. In fact, exp 1 shows that complete reminders during wake strengthen memory, and incomplete reminders are only necessary (in interaction with interference) to weaken them. Therefore, why shouldn't complete reminders strengthen memories during sleep as they do during wake? Why is labilization, which is the unique feature of memory reconsolidation during wake, critical for memory consolidation (i.e., strengthening but not updating) during sleep? Why should sleep-related consolidation rely on the mechanism involved in reconsolidation? This is the critical theoretical claim at the base of the authors' hypothesis and I'm not convinced I understand the logic they used in developing this hypothesis.

B. The striking finding in study 2 is the difference between the results for the Rc condition for exp 1 vs exp 2&3. The discussion considers the time delay between reactivation and test as the most critical difference between experiments, but there are several differences including circadian effects, tiredness and the temporal proximity between training/reactivation and interference and between training and test. These limitations are briefly mentioned in the results section, but should be discussed in the discussion section as well.

C. Throughout the paper, a 40-minute nap setting is mentioned. However, I am not sure how this duration is defined. For example, in one of the experimental conditions (Ri, exp 3) the total sleep time is 74 minutes on average. This is likely due to the fact that N2 was also longer, but the rules defining when the naps were terminated are not mentioned in detail. Also, the number of times each cue was presented is not explicitly stated.

D. The authors claim (lines 685-690) that late wakefulness may be beneficial for the Ri-related benefit. This assumes that two separate active processes maintain memory, one during early sleep (supporting both Ri and Rc) and one during late sleep/wake (supporting Ri). However, it is just as likely that the difference between Ri and Rc during the later part concerns the (passive) decay of the memory in the Rc condition and not the (active) promotion of the Ri memory. Assuming the relative memory benefit in the Ri condition is established already after 40 minutes of sleep and remains unchanged thereafter, it seems even more likely that the difference recorded in exp 2&3 is best explained by a decay in Rc. If this is true, the claim that a two-stage model can be extended to include wake consolidation may be less convincing.

Minor comments:

1. Line 31 - the phrase "based on a mismatch induced by incomplete reminders" may not be clear to readers who have not read the full paper. The authors should clarify, as they do later in the paper, why an Ri creates a mismatch.
2. Line 114 - how did the authors define "good physical and mental health"? This is a rather vague

term.

3. Lines 203-210 - the authors should state in the description of exp 2 that sleep was followed by interference and test, as they do for exp 1 and exp 3 before and after this paragraph.
4. Line 205 - the term "40 min sleep reactivation period" may mislead readers into thinking all 40 minutes were dedicated to reactivation.
5. Line 236 - the authors should mention that these 1500 ms also included the sounds.
6. Figure 1 - for readability, it would be helpful to have the time arrow (and the screen shots) go from top left to bottom right.
7. Were the headphones in-ear or not? How did the authors confirm that they did not shift from participants' ears during sleep?
8. Line 306 - the sounds were stopped and then restarted if the participants woke up, but in line 133 it is stated that participants that woke up were excluded. Which is correct?
9. I suggest renaming the experiments in studies 1 and 2 to avoid confusion (i.e., numbering them from 1 to 4, where exp 2-4 are associated with study 2). Often there are references to exp 1 and it may be confusing to understand which experiment is referred to (exp 1 in study 1 or exp 1 in study 2).
10. Lines 338-350 - this procedure only applies to study 2 and this should be explicitly noted. For example, the words "all experiments" in line 348 actually only refers to the ones in study 2.
11. Lines 315-317 - were the words used as part of the interference task also semantically related to the sounds? Were they equally related as were the words in the original task? Were these words also made of three syllables? Was the first syllable different than the one used in the original task?
12. Several comparisons between two groups were reported using F statistics and not T statistics (e.g., lines 399, 405, 408, 409).
13. Figure 3 - the colors/patterns used in this figure were confusing. First, the pattern for "wake" in panel A is very similar to the one for "sleep", making it hard to see the difference between exp 2 and 3. Second, the two panels use the same patterns and colors to signify different things, complicating the interpretation (e.g., the color used for Rc in B is used to indicate "Training" in A).
14. Several p-values are used without reporting the statistics (e.g., line 526).
15. If I understand the analysis for the "heard/not-heard" data correctly, the average across subjects for the NR condition was considered as chance and the data for the other conditions was compared to this absolute data. However, the between-subject average should not be taken as a constant representing chance. It is also a random variable, which is characterized by some variability in measurement. It would make more sense to use an independent-measures t-test between the NR and each of the Ri/Rc groups. This is true for all three experiments of study 2.
16. Some details of the cluster analysis are missing. For example, it is unclear what the authors mean by the "theta and spindle bands". Which frequency range and time range emerged as significant? What was the significance level of these clusters? More information on the method and results should be reported. Showing the significant cluster in the figure may also be helpful.
17. It is puzzling that the sound preceding the words did not, by itself, elicit a theta response. Targeted memory reactivation studies using sounds (and not words) have shown theta increases, raising the possibility that the sound in this study may have been presented too softly. Alternatively, this lack of an effect could be due to the spectrometer not being locked to the sound (which has a jittered onset), but to the word. Either way, this issue should be addressed by the authors.
18. If I understand correctly, the same interference task was used for two very different purposes in study 1 and 2: in study 1 it was used as part of a reconsolidation paradigm to modify existing memories; in study 2 the same protocol was used not to modify memory, but rather to enhance the contrast between conditions in the subsequent test. If this is correct, it is worth mentioning, because these very different purposes (to edit memory traces in study 1 and to enhance signal to noise in study 2) are somewhat confusing.
19. Line 639 - it may be justified to cite a recent paper suggesting this very model (Lewis and Bendor, *Curr Bio*, 2019).
20. The authors relate their findings to those of Schreiner & colleagues (2015) and explain the lack of a cueing effect in that paper as further evidence for partial cueing being more beneficial. However, that study also showed that presenting a tone after the partial cue inhibits consolidation.

It seems as though it is not the partial aspect of the cue, but rather the interference between the different parts of the cue, that best explain the lack of a TMR effect. Therefore, it is less likely that the results reflect a partial vs full cuing difference like the one in the current manuscript.

21. The authors mention that MMN still persists (partly) during sleep and that this supports their hypothesis that prediction errors play a part in consolidation. However, MMN has mostly been observed in light stages of sleep and in REM sleep (e.g., Scultholpe et al., 2009). Therefore, this evidence does not support the suggested model, which concentrates on consolidation during NREM. As a side note, the citation in line 654 appears to be wrong.

Eitan Schechtman

Reviewer #2 (Remarks to the Author):

This is an interesting and well written paper describing a well designed study that has been carefully analysed. The authors ask whether TMR reminder cues are differentially effective if they are complete or partial cues for the learned information. This questions arose out of a literature showing that such complete/partial cuing does lead to differential impacts in wake, with incomplete reminders more effective at reactivating memories. In a series of experiments they first establish that only incomplete reminders labilized memories during wake by showing that interference following these reminders (but not incomplete reminders) had a significantly detrimental effect. They then examined TMR reminders in a 40 minute sleep followed by interference and testing after 1) no time 2) ~ 6 hours of sleep, 3) 6 hours of wake. Results showed that both complete and incomplete reminders protected memory against interference if tested immediately. If tested remotely, only incomplete reminders offered protection. This was significant in the sleep group, but with a strong trend of .07 in the wake group (3). Looking at the electrophysiology, both complete and incomplete reminders induced theta and spindles, with spindles being stronger in the complete reminder group.

I think this paper is solid, and have no major technical points. However I have a few comments and queries/suggestions.

Minor:

- 1) The experimental paradigm drawing in fig 3 A is confusing. For instance, it is not clear that the forward stripy box is sleep while backwards stripey is wake. It really took me awhile to figure this out. Please rework this figure, label everything better, and try to make it clearer.
- 2) When describing the exploratory analysis please give more detail about the factor in our ANOVA (was it all 3 experiments)?
- 3) Figure 4: it would be good to see a plot of the comparison between complete and incomplete – e.g. the spindles were significantly different and it would be nice to see this represented.
- 4) Were there any correlations between spindle/theta power and TMR effects? This would be particularly interesting given the difference in response to complete/incomplete cues.
- 5) The point about wake being equivalent to REM is important and could have more air time. Were there correlations between behavioural findings and sleep stage minutes or percentages?

Reviewer #3 (Remarks to the Author):

Reactivation during sleep with incomplete (but not complete) reminder cues stabilizes memory for the long-term.

Forcato et al.

In this study Forcato and colleagues investigated whether providing reminder cues during sleep is

more efficient when presenting incomplete as compared to complete reminders, as it has been shown during wakefulness in the context of reconsolidation. In a first step the authors show that only incomplete reminder cues during wakefulness labilize memories. In contrast, when presenting both complete and incomplete reminder cues during 40 minutes of NREM sleep, both stimulus categories were capable of stabilizing later memory performance. In a next step the authors extended the retention interval to 7 hours (in addition to the initial 40 minutes of sleep). These retention intervals were either filled with wakefulness or sleep. Interestingly by lengthening the retention interval and irrespective of the state of wakefulness or sleep, complete reminders lost their potential to bias memory consolidation, while incomplete reminders were still capable of stabilizing memories. Overall, the reported study and associated results are timely and quite interesting. Unfortunately I am rather concerned with some features of the manuscript, which I hope the authors may address.

Major comments:

Comment 1: As far as I understood, the authors propose that the underlying process driving their results (thus the effectiveness of incomplete reminders to labilize memories during wake and stabilize memories during sleep) is a prediction error / mismatch detection mechanism. To induce such mismatch perception during wakefulness, participants of study 1 attended on the day after the learning session a reactivation session. Here they received reminders of the learning task (either complete reminder or incomplete reminder). For the incomplete reminder condition only the first syllable of each word which were associated before with a sound was presented. This procedure was meant to induce a prediction error and consequently reconsolidation in the participants, assuming that they would have expected to hear the whole word instead of a cropped version.

Unfortunately, I have some struggles to follow this reasoning. As far as I can see, a very similar procedure was already administered during memory retrieval the day before. Here, participants were presented with sounds plus the first syllable of the associated words (thus a incomplete reminder) and had to vocalize the missing part of each word. Thus, participants were already quite familiar with being presented with incomplete reminders, rendering further expectation violations rather unlikely.

Apart from that even if they were not familiar with the procedure, it seems likely that presenting incomplete reminders could lead to a mismatch during the first couple of trials. But could one really assume that there is still such a prediction error after let's say 20 trials? I would be quite interested about the thoughts of the authors concerning this topic.

Comment 2: My second concern is quite related to the one above. The fact that only incomplete reminders presented during sleep showed consistent effects on memory stabilisation led the authors to suggest that also here some sort of mismatch detections should be active during sleep, with the prediction error tagging memories for late consolidation. While it is known that prediction errors might be elicited at least to some extend during sleep (manifesting in distinct EPRs as discussed by the authors), I just don't see why the memory system should be geared towards mismatch detection during sleep. Or in other words the authors present no data in clear favour of this explanation.

Thus, if the authors believe that a mismatch detection process is driving their results, I wonder why they did not check in their sleep data whether they can find any difference between the ERPs related to complete and incomplete reminders. This would tremendously help their argumentation.

Comment 3: I have some concerns with regards to the EEG analyses and their interpretation.

- The authors report that EEG segments containing the reminder cues were contrasted against comparable time-windows of the no-reminder control group. They do not give any more detail how that was achieved, even though it would help to understand the shown figure.
- The authors report that each reminder (30 in total) was presented once, leading to a rather low

stimulus number. Given that trials were rejected during pre-processing, it would be good to know how many of them entered the final analysis.

- In my view the biggest problem concerning the EEG results is that it is entirely unclear whether they comprise any memory-relatedness. The authors did not present any memory unrelated control stimuli during sleep, which could have served as meaningful contrast / control condition. And I assume that the low trial number hinders them from contrasting their stimuli according to a subsequent memory rationale. Thus, I fear that the main activation which can be seen in Figure 4 is in main parts related to the processing of auditory stimuli during sleep.

The result that complete reminders tended to elicit stronger activity in the sleep spindle range might just be related to the fact that those stimuli were longer in duration as compared to incomplete reminders. This would also fit to the known sleep-preserving role of sleep spindles (e.g. Dang -Vu et al., 2010). Thus stimuli of longer duration might just have a generally higher capability to wake participants up and thus elicit preferentially sleep spindles, which in turn protect the sleeping brain.

- I was wondering why the EEG analysis was time-locked to the word cues and not the preceding sounds. Wouldn't the authors assume that hippocampal pattern completion and hence reactivation processes should already (and preferentially) be elicited by these initial cues?

In sum I don't see that the EEG analyses and results in the present form help to understand the behavioural effects reported by the authors by any means.

Minor comment: The authors state in the introduction that ...the mechanisms underlying memory stabilization upon cued reactivation in the sleep state are largely unknown... Maybe the authors would want to include the work of Bendor & Wilson (2012) as well as recent findings by Rothschild and colleagues (2017), which sheds some light on the neural processes associated with targeted memory reactivation.

Response to Reviewers

We would like to thank the Reviewers for the very fruitful, constructive and helpful comments. In the following we address their concerns point by point.

Reviewer #1.

Forcato and colleagues explore the parallels between memory reconsolidation during wake and consolidation during NREM sleep by running multiple experiments exploring the benefits of complete and incomplete reminders to memory during wake and sleep. Drawing from the literature on targeted memory reactivation, they conclude that incomplete - but not complete - reminders during sleep lead to prolonged benefits to memory. The manuscript is well written and explores a previously unaddressed question using appropriate methods and analyses. There are some minor issues regarding the theoretical background and interpretation that need to be addressed before it should be accepted for publication in this journal.

Major issues:

A. The main hypothesis in this manuscript is that the cognitive mechanism involved in reconsolidation during wake may similarly affect consolidation during sleep. In reconsolidation paradigms, partial cuing during wake labilizes memories and allows updating. However, as the authors themselves show, reconsolidation using partial cues is not necessary for memory strengthening. In fact, exp 1 shows that complete reminders during wake strengthen memory, and incomplete reminders are only necessary (in interaction with interference) to weaken them.

Therefore, why shouldn't complete reminders strengthen memories during sleep as they do during wake? Why is labilization, which is the unique feature of memory reconsolidation during wake, critical for memory consolidation (i.e., strengthening but not updating) during sleep? Why should sleep-related consolidation rely on the mechanism involved in reconsolidation? This is the critical theoretical claim at the base of the authors' hypothesis and I'm not convinced I understand the logic they used in developing this hypothesis.

We thank the reviewer for this comment and the opportunity to elaborate on our reasoning in more detail. In fact, in our previous studies, we did not observe a strengthening of the memories with complete reminders. Memory strengthening was

only observed if we labilized a consolidated memory by presenting an incomplete reminder and at least one more incomplete reminder was presented inside the time window of the labilization/reconsolidation period (Forcato et al., 2011; 2013). This strengthening effect was not observed if a complete reminder was presented. Thus, we concluded that in order to observe memory strengthening in the wake state, at least two incomplete reminders should be presented and that both of them should induce memory labilization and not just retrieve the memory. Interestingly, previous sleep cueing studies almost exclusively applied cues that can be classified as incomplete reminders (Schouten et al., 2017; Klinzing & Diekelmann 2019). For instance, odor cues can be considered incomplete reminders because they serve as context cues for the associated learning material. Sound cues are typically associated with other contents like words or pictures and can therefore be considered incomplete reminders as well. Therefore, we hypothesized that in order to induce memory strengthening during sleep the cues should act as incomplete reminders.

Importantly, study 1 in our present manuscript does not allow for any conclusions with regard to memory strengthening because we did not include a group without reactivation. In order to show memory strengthening, the reminder group(s) would be expected to perform better than a no-reactivation control. Study 1 was specifically designed to test for memory labilization by introducing an interfering task after reactivation. We show that the complete reminder did not labilize the memory, since it was not impaired by subsequent interference learning. Yet, we cannot say whether the memory was strengthened or simply unchanged. Based on our previous findings outlined above, we believe that the latter is true, i.e. the memory remains unchanged (and is not strengthened) upon cueing with complete reminders.

We agree that the idea that memory consolidation may rely on mechanisms of labilization and updating similar to wake reconsolidation seems counterintuitive at first. However, modern theories conceptualize consolidation not as a simple (passive) process of strengthening of single memory traces but as a complex active process of restructuring and integration of new contents into pre-existing networks, even including processes of abstraction and schema extraction. For such an integrative process, labilization and updating may be essential for a complex re-configuration and adaptation of the whole network. This hypothesis could arguably be wrong but we still believe it is worth testing and we have tried to provide a clearer explanation of our reasoning in the introduction:

Lines 61-69: "Incomplete reminders have been shown to be more effective than complete reminders in triggering the reactivation and updating of memories when single incomplete reminders were presented in the wake state, followed by interference

learning (or similar amnesic agents) within the labilization window of about 6 hours^{5,15}. Incomplete reminder cues were also more effective in inducing memory strengthening when, instead of interference learning, a second incomplete reminder was presented while the memory was still in a labile state⁶⁻¹⁶. However, this strengthening effect was not observed when a complete reminder was presented in this time window of lability⁶.”

Lines 89-114: “Interestingly, previous sleep cueing studies almost exclusively applied cues that can be classified as incomplete reminders⁴⁰⁻⁴². For instance, odor cues can be considered incomplete reminders because they serve as context cues for the associated learning material^{33,38}. Sound cues are typically associated with other contents like words or pictures and can therefore be considered incomplete reminders as well. Memory strengthening in sleep reactivation studies is frequently observed after only a single presentation of the incomplete reminder, while in wake reactivation studies a second incomplete reminder is necessary to induce memory strengthening after the first incomplete reminder has triggered labilization. This difference could be explained by the fact that reminder presentation in sleep studies almost always takes place shortly after learning, i.e. while the memory is still in a labile state and not already consolidated as in typical wake reactivation studies. However, it is presently unclear whether cueing with incomplete reminders is more effective than cueing with complete reminders during sleep, similar to the wake state. Memory consolidation during sleep has recently been conceptualized as a complex process involving the restructuring and integration of new contents into pre-existing networks, including processes of abstraction and schema extraction^{43,44}. For such an integrative process, labilization and updating may be essential for a complex re-configuration and adaptation of the whole network. Considering that labilization and updating during wakefulness depend on incomplete reminder presentation, we expect cueing with incomplete reminders during sleep to be more effective than with complete reminders. Some evidence suggests that at least a rudimentary form of mismatch detection, which is assumed to underlie incomplete reminder effects during wakefulness, is still in place during sleep⁴⁵.”

We have also mentioned in the results section that we cannot draw any conclusions with regard to the memory strengthening effect from study 1 (lines 223-228):

“These results show that during wakefulness only the incomplete reminder labilizes the sound-word associations making them vulnerable for disruption by the interference task, whereas the complete reminder does not labilize the underlying memory traces. Note that this study does not allow for any conclusions regarding memory strengthening, which would require comparing the reminder groups to a group without reminder presentation.”

B. The striking finding in study 2 is the difference between the results for the Rc condition for exp 1 vs exp 2&3. The discussion considers the time delay between reactivation and test as the most critical difference between experiments, but there are several differences including circadian effects, tiredness and the temporal proximity between training/reactivation and interference and between training and test. These limitations are briefly mentioned in the results section, but should be discussed in the discussion section as well.

We agree with the reviewer and highlighted these issues in the discussion as follows (lines 668-677):

“Generally, the comparison between experiments should be interpreted with caution because there are several fundamental differences between experiments including circadian effects, tiredness, sleep length, and temporal proximity between training, reactivation and interference/testing, all of which are known to affect memory performance. Particularly relevant seems to be the difference in temporal proximity between reactivation and interference/testing (~30 min in Exp. 2 vs. ~8h in Exp. 3 and 4), differences in total sleep time (~47 min in Exp. 2, ~465 min in Exp. 3, ~67 min in Exp. 4) as well as different states of tiredness at testing (rested in Exp. 2 and 3 vs. sleep deprived in Exp. 4), adding further limitations to the study.”

C. Throughout the paper, a 40-minute nap setting is mentioned. However, I am not sure how this duration is defined. For example, in one of the experimental conditions (Ri, exp 3) the total sleep time is 74 minutes on average. This is likely due to the fact that N2 was also longer, but the rules defining when the naps were terminated are not mentioned in detail. Also, the number of times each cue was presented is not explicitly stated.

We thank the reviewer for this comment. When we performed the experiment, we counted 40 min from the first spindle detected by the experimenter online and when that time was accomplished we woke up the participants in the 40 min and 7 h groups. For the groups that received the reactivation during SWS, the experimenter had to wait 10 min after the subjects reached stable SWS before starting the reactivation. Thus, in some cases the experimenter had to wait longer until the participant reached stable SWS. Moreover, the reactivation was paused whenever signs of arousal or changes in sleep stage were detected, and resumed upon stable SWS, likewise adding some variance in overall sleep time. Each cue was presented once during reactivation, that is, all participants received 30 reactivation trials. However, for the EEG reactivation

analysis after artifact rejection, overall 16 reactivation trials were excluded for Rc, 10 trials for Ri and 12 for NR.

We included all these details in lines 909-923:

“In all experiments, reminders were presented only in SWS during the first ~40 min of sleep. The 40 min sleep duration was counted from the first spindle or K-complex detected by the experimenter online. For groups that received the reactivation during SWS, the experimenter had to wait 10 min after the subjects reached stable SWS before starting the reactivation. Thus, in some cases the experimenter had to wait longer until the participant reached stable SWS. Moreover, the reactivation was paused whenever signs of arousal or changes in sleep stage were detected, and resumed upon stable SWS was detected again, additionally adding some variance in individual sleep time. Each cue was presented once during reactivation, that is, all participants received 30 reactivation trials. The entire reactivation procedure took about 5 minutes and 45 seconds. Participants were awakened after the reactivation procedure (i.e. 30 reactivation trials) was completed and they had slept for at least 40 min (Exp. 2 and 4) or 8 h (Exp. 3). After awakening, electrodes were removed.”

Lines 965-969: “Trials containing artefacts in any of the channels were removed using automatic and visual rejection. After data preprocessing, EEG data from 108 subjects remained for further analyses, overall 16 reactivation trials were excluded for Rc, 10 trials for Ri and 12 trials for NR.”

D. The authors claim (lines 685-690) that late wakefulness may be beneficial for the Ri-related benefit. This assumes that two separate active processes maintain memory, one during early sleep (supporting both Ri and Rc) and one during late sleep/wake (supporting Ri). However, it is just as likely that the difference between Ri and Rc during the later part concerns the (passive) decay of the memory in the Rc condition and not the (active) promotion of the Ri memory. Assuming the relative memory benefit in the Ri condition is established already after 40 minutes of sleep and remains unchanged thereafter, it seems even more likely that the difference recorded in exp 2&3 is best explained by a decay in Rc. If this is true, the claim that a two-stage model can be extended to include wake consolidation may be less convincing.

This is an interesting point and we agree with the reviewer. This is indeed an alternative possibility and we included this in the new version of the manuscript in the discussion section as follows (lines 659-667):

“Our data extend this view in proposing that late periods of wakefulness may be similarly effective than REM sleep in supporting memory stabilization for the long-term.

However, assuming that the memory benefit in the incomplete reminder condition is established already after 40 minutes of sleep and remains unchanged thereafter, the difference between reminder types could be alternatively explained by a memory decay for the complete reminder condition during the extended retention interval. These two alternative explanations cannot be discerned with the present experimental design and should be subject to further investigation.”

Minor comments:

1. Line 31 - the phrase "based on a mismatch induced by incomplete reminders" may not be clear to readers who have not read the full paper. The authors should clarify, as they do later in the paper, why an Ri creates a mismatch.

Thank you, we clarified the meaning of mismatch (lines 28-30):

“In reconsolidation, incomplete reminders are more effective in reactivating memories than complete reminders by inducing a mismatch, i.e. a discrepancy between expected and actual events.”

2. Line 114 - how did the authors define "good physical and mental health"? This is a rather vague term.

We clarified this issue and modified the sentence as follows (lines 719-724):

“None of the participants reported being sick during the experiment, to have any mood disorder or any anxiety disorder, any history of neurological diseases, did not take any medication at the time of the experiments, did not suffer from any sleep disorders, had not been on night shift and did not have any major sleep disturbances during the 6 weeks prior to the experiments.”

3. Lines 203-210 - the authors should state in the description of exp 2 that sleep was followed by interference and test, as they do for exp 1 and exp 3 before and after this paragraph.

In the new version we included that sleep was followed by the interference task and testing session as follows (lines 823-826):

“Importantly, no further reminders were presented during the additional 7 h sleep interval. 30 min after being awakened, the participants received the interference task and 30 min later they were tested on the original task.”

4. Line 205 - the term "40 min sleep reactivation period" may mislead readers into thinking all 40 minutes were dedicated to reactivation.

Yes, we agree. We modified the sentence as follows (lines 820-824):

“In this experiment, the procedures were basically identical to Experiment 2, except that participants were left sleeping for additional 7 hours during the experimental night after the first 40 min sleep period during which the reactivation occurred. Importantly, no further reminders were presented during the additional 7 h sleep interval.”

5. Line 236 - the authors should mention that these 1500 ms also included the sounds.

We thank the reviewer for this observation. We modified the sentence as follows (lines 154-158):

“In the reactivation session, each sound was first presented alone for on average 2,900 ms. The sound then continued and the first syllable of each word (incomplete reminder) or the entire word (complete reminder) was presented once in addition to the sound for 1,500 ms”

6. Figure 1 - for readability, it would be helpful to have the time arrow (and the screen shots) go from top left to bottom right.

We modified figure 1 accordingly.

Figure 1. Memory task. During the training session, participants learned 30 sound-word associations. For each association, the sound was presented first followed by the presentation of the sound plus the word written on the screen and spoken aloud via headphones. The next association appeared after a 4,000 ms break. After all 30 associations had been presented once (learning), participants completed an immediate cued recall test. For each association, the sound and the first syllable of the associated word were presented and participants had to say the associated word aloud upon appearance of an image of a microphone on the screen, while the sound continued *during the entire period*. *Independent of the participant's response, the correct answer was then displayed written on the screen and via headphones.* In the reactivation session, each sound was first presented alone for on average 2,900 ms. The sound then continued and the first syllable of each word (incomplete reminder) or the entire word (complete reminder) was presented once in addition to the sound for 1,500 ms. Although in Study 1, participants were instructed to say the associated word aloud every time the microphone appeared on the screen, the microphone never actually appeared and each trial was interrupted so that participants never had the chance to give a response. In Study 2, exactly the same reactivation procedure was applied during sleep, except for the instruction to respond. In the testing session, participants were presented with the sound of each association and had to say the associated word aloud. Musical notes indicate presentation of the sound.

7. Were the headphones in-ear or not? How did the authors confirm that they did not shift from participants' ears during sleep?

We have clarified this point as follows (lines 801-804):

“Participants were presented with white noise (43 dB) via in-ear headphones from the time they went to bed until they were awakened. The in-ear headphones were attached to the ear with a hypoallergenic tape and it was ensured that the headphones were still in place upon waking up.”

8. Line 306 - the sounds were stopped and then restarted if the participants woke up, but in line 133 it is stated that participants that woke up were excluded. Which is correct?

Thank you for noticing this mistake. The sounds were stopped if the participants showed arousal or changes in sleep stage, and participants were excluded if they woke up. We corrected the error as follows (lines 915-918):

“Moreover, the reactivation was paused whenever signs of arousal or changes in sleep stage were detected, and resumed upon stable SWS was detected again, additionally *adding some variance in individual sleep time.*”

9. I suggest renaming the experiments in studies 1 and 2 to avoid confusion (i.e., numbering them from 1 to 4, where exp 2-4 are associated with study 2). Often there are references to exp 1 and it may be confusing to understand which experiment is referred to (exp 1 in study 1 or exp 1 in study 2).

We agree with the reviewer and changed the numbering of the experiments as suggested throughout the manuscript.

10. Lines 338-350 - this procedure only applies to study 2 and this should be explicitly noted. For example, the words "all experiments" in line 348 actually only refers to the ones in study 2.

We agree with the reviewer. We modified the whole section as follows (lines 957-980):

“Reactivation EEG data processing

All experiments in Study 2 diverged only after the reactivation phase. Therefore, electrophysiological data for the reactivation period of the three experiments were pooled for each condition, resulting in three datasets: complete reminder (Rc),

incomplete reminder (Ri) and no reminder (NR). EEG data of each reactivation event was cut into 12-sec trials (-7 to 5 sec, with $t = 0$ sec referring to cue onset). Because the reminder conditions are identical during the sound presentation interval and only differ upon presentation of the cue (syllable/word), the analyses were time-locked to cue onset. Trials containing artefacts in any of the channels were removed using automatic and visual rejection. After data preprocessing, EEG data from 108 subjects remained for further analyses, overall 16 reactivation trials were excluded for Rc, 10 trials for Ri and 12 trials for NR. Time-frequency analysis was performed (10-cycle Morlet wavelets) to calculate the power for the average of the electrodes C3 and C4 relative to a baseline of one second right before sound onset (-4 and -3 sec). This processing resulted in the relative power of each frequency at each time point for Rc, Ri and NR independently. All reactivations were aligned to cue onset ($t=0$ sec). Significant responses upon the sound and cue presentation in the Rc and Ri conditions were compared to the NR condition as well as between both reminder conditions. *Responses upon the sound presentation are called “sound-evoked” responses and responses upon the cue presentation are called “cue-evoked” responses. Different significant clusters were detected, according to the classical band frequencies for slow spindles (10-12 Hz), fast spindles (12-16 Hz) and theta (4-8 Hz).”*

11. Lines 315-317 - were the words used as part of the interference task also semantically related to the sounds? Were they equally related as were the words in the original task? Were these words also made of three syllables? Was the first syllable different than the one used in the original task?

The words in the interference task were also semantically related to the sounds. Before running study 1, we performed a pilot study, in which participants ranked different associations (from 1 = no semantic association to 10 = high semantic association), and we chose two words for each sound that had the same mean level of association. The first syllables of the words were different than in the original task, but all words also consisted of three syllables. We clarified as follows (lines 932-936):

“The interference task consisted of the same sounds from the training session but those sounds were now associated with other words. These words were also semantically related to the sounds and likewise consisted of three syllables, but with a different first syllable than the words of the original task. The procedure was the same as for the training session.”

12. Several comparisons between two groups were reported using F statistics and not T statistics (e.g., lines 399, 405, 408, 409).

Yes, for significant interactions in the respective ANOVAs we performed post-hoc simple effects analyses (with F statistics) instead of t-Tests as detailed in lines 1018-1019.

13. Figure 3 - the colors/patterns used in this figure were confusing. First, the pattern for "wake" in panel A is very similar to the one for "sleep", making it hard to see the difference between exp 2 and 3. Second, the two panels use the same patterns and colors to signify different things, complicating the interpretation (e.g., the color used for Rc in B is used to indicate "Training" in A).

We thank the reviewer for this observation. To avoid any confusion, we decided to use different colors in a new version of the figure.

Figure 3. Memory stabilization by different reminder cues during sleep in Study 2. (a) In three experiments, all participants were trained in the evening before a 40-min sleep period during which the complete reminder (Rc groups), incomplete reminder (Ri groups) or no reminder (NR groups) was presented. In Exp. 2, participants were awakened after the 40-min sleep period, learned the interference task 30 min later and were tested on the original task another 30 min later. In Exp. 3, procedures were exactly the same for training and reminder presentation, but participants were allowed to continue sleeping after reminder presentation and were awakened in the morning, with interference learning and testing taking place thereafter. In Exp. 4, procedures were also identical for training and reminder presentation, but participants were

awakened after the 40 min sleep period and remained awake for another 7 hours, with interference learning and testing taking place thereafter. (b) In Exp. 2 (40min), both reminder groups performed better than the no reminder group, with no difference between the complete and incomplete reminder groups (left panel). In Exp. 3 (8h), the incomplete reminder group performed better than the complete reminder and no reminder groups, with no difference between the complete and no reminder groups (middle panel). In Exp. 4 (40min-7h), the incomplete reminder group performed better than the complete reminder group and showed a trend toward better performance than the no reminder group, with no difference between the complete and no reminder groups (right panel). Memory change represents the number of correct responses at testing minus training, i.e. indicating forgetting. Means \pm SEM are shown. * $P < 0.05$; # $P = 0.072$.

14. Several p-values are used without reporting the statistics (e.g., line 526).

We added the statistics throughout the manuscript.

15. If I understand the analysis for the "heard/not-heard" data correctly, the average across subjects for the NR condition was considered as chance and the data for the other conditions was compared to this absolute data. However, the between-subject average should not be taken as a constant representing chance. It is also a random variable, which is characterized by some variability in measurement. It would make more sense to use an independent-measures t-test between the NR and each of the Ri/Rc groups. This is true for all three experiments of study 2.

We agree with the reviewer and removed the chance comparisons. However, as we reported the ANOVA across all groups for each experiment showing no significant differences in the main ANOVA, we decided not to include further t-test.

16. Some details of the cluster analysis are missing. For example, it is unclear what the authors mean by the "theta and spindle bands". Which frequency range and time range emerged as significant? What was the significance level of these clusters? More information on the method and results should be reported. Showing the significant cluster in the figure may also be helpful.

We have now included more information on the methods of the cluster analysis and hope that the procedures are clearer now. By "theta and spindle bands" we referred to the classical band frequencies for fast spindles (12-16 Hz) and theta (4-8 Hz). In Figure 4, we referred to the clusters compatible with these bands. Moreover, we named the clusters according to the time point where they were detected ($t = 0$ sec aligned to the

cue onset), therefore we called them “sound-evoked” and “cue-evoked” clusters. Data were masked and contrasted using comparable time windows (e.g. 1 sec windows) between the no-reminder and complete reminder conditions (Fig. 4a), same for the no-reminder and incomplete reminder condition (Fig. 4b). Therefore, the non-shaded areas in these figures show those areas in which differences between both conditions are significant. The P values corresponding to each condition and cluster are reported in the Figure 4 legend and we have now added them to the text of the manuscript. Nevertheless, the extent of such clusters should generally be interpreted with caution. The extent of frequency bands and time ranges of significant clusters cannot be interpreted when using cluster-based permutation testing.

([http://www.fieldtriptoolbox.org/faq/how not to interpret results from a cluster-based permutation test/](http://www.fieldtriptoolbox.org/faq/how_not_to_interpret_results_from_a_cluster-based_permutation_test/)).

We have modified the manuscript as follows lines 427-455:

“Electrophysiological analysis of cued reactivation during sleep

To test for differences in brain oscillation patterns upon reactivation with the different reminder types, time-frequency analyses were performed for each condition: Rc (complete reminder), Ri (incomplete reminder), and NR (no reminder). EEG data for the reactivation period were contrasted using equivalent time windows between the NR and Rc conditions (Fig. 4a), between the NR and Ri conditions (Fig. 4b), as well as between the Rc and Ri conditions (Fig. 4c). Following the presentation of the cue, both complete and incomplete reminder cues (i.e. words and syllables, respectively) evoked responses in the theta band (4-8 Hz) and the fast spindle band (12-16 Hz, all cluster-level $P < 0.001$) as compared to the no reminder condition. When both reminder conditions were directly compared, cue-evoked power increases in the slow spindle band (10-12 Hz) were stronger in the complete reminder compared to the incomplete reminder condition (Fig. 4c, $P = 0.025$), whereas power increases in the theta band (4-8 Hz) did not differ significantly ($P = 0.15$). Regarding responses evoked by the presentation of the sound (which was identical for both reminder conditions), significant power increases were detected in the fast spindle band in both reminder conditions (12-16 Hz, Rc vs. NR: Fig. 4a $P = 0.032$, Ri vs. NR: Fig. 4b $P = 0.029$); yet despite descriptive increases in theta power in response to the sound, there were no significant clusters spanning that frequency range (Rc vs. NR: $P = 0.40$, Ri vs. NR: $P = 0.95$). Interindividual differences and variability in the sound recordings may have introduced variance obscuring sound-induced power changes, which have typically been observed in previous studies using sounds as reactivation cues⁴⁶. However, a genuinely attenuated theta response would also be in line with previous research, suggesting that additional sounds immediately following a reminder cue can disrupt processing of that

initial reminder⁴⁷⁻⁴⁸. Long-lasting auditory stimulation like the sounds used in the present study may yield a similar effect.”

Lines 957-980: “Reactivation EEG data processing

All experiments in Study 2 diverged only after the reactivation phase. Therefore, electrophysiological data for the reactivation period of the three experiments were pooled for each condition, resulting in three datasets: complete reminder (Rc), incomplete reminder (Ri) and no reminder (NR). EEG data of each reactivation event was cut into 12-sec trials (-7 to 5 sec, with $t = 0$ sec referring to cue onset). Because the reminder conditions are identical during the sound presentation interval and only differ upon presentation of the cue (syllable/word), the analyses were time-locked to cue onset. Trials containing artefacts in any of the channels were removed using automatic and visual rejection. After data preprocessing, EEG data from 108 subjects remained for further analyses, overall 16 reactivation trials were excluded for Rc, 10 trials for Ri and 12 trials for NR. Time-frequency analysis was performed (10-cycle Morlet wavelets) to calculate the power for the average of the electrodes C3 and C4 relative to a baseline of one second right before sound onset (-4 and -3 sec). This processing resulted in the relative power of each frequency at each time point for Rc, Ri and NR independently. All reactivations were aligned to cue onset ($t=0$ sec). Significant responses upon the sound and cue presentation in the Rc and Ri conditions were compared to the NR condition as well as between both reminder conditions. *Responses upon the sound presentation are called “sound-evoked” responses and responses upon the cue presentation are called “cue-evoked” responses. Different significant clusters were detected, according to the classical band frequencies for slow spindles (10-12 Hz), fast spindles (12-16 Hz) and theta (4-8 Hz).”*

Lines 1033-1040: “*To assess the statistical differences between the EEG response to the reactivation in each condition, the time-frequency values previously calculated (see Reactivation EEG processing) were compared using sample-level two-tailed independent-samples t-tests followed by a non-parametric cluster-permutation procedure to correct for multiple comparisons⁸⁰ (5000 permutations), as implemented in the open-source toolbox FieldTrip⁷⁹. All EEG data processing was done using Matlab 2017a (MathWorks Inc, Natick, Massachusetts, USA).”*

17. It is puzzling that the sound preceding the words did not, by itself, elicit a theta response. Targeted memory reactivation studies using sounds (and not words) have shown theta increases, raising the possibility that the sound in this study may have been presented too softly. Alternatively, this lack of an effect could be due to the spectrometer not being locked to the sound (which has a jittered onset), but to the word. Either way, this issue should be addressed by the authors.

We agree with this observation. In the new version of the manuscript we have included a discussion of this issue as follows (lines 442-455):

“Regarding responses evoked by the presentation of the sound (which was identical for both reminder conditions), significant power increases were detected in the fast spindle band in both reminder conditions (12-16 Hz, Rc vs. NR: Fig. 4a $P = 0.032$, Ri vs. NR: Fig. 4b $P = 0.029$); yet despite descriptive increases in theta power in response to the sound, there were no significant clusters spanning that frequency range (Rc vs. NR: $P = 0.40$, Ri vs. NR: $P = 0.95$). Interindividual differences and variability in the sound recordings may have introduced variance obscuring sound-induced power changes, which have typically been observed in previous studies using sounds as reactivation cues⁴⁶. However, a genuinely attenuated theta response would also be in line with previous research, suggesting that additional sounds immediately following a reminder cue can disrupt processing of that initial reminder⁴⁷⁻⁴⁸. Long-lasting auditory stimulation like the sounds used in the present study may yield a similar effect.”

18. If I understand correctly, the same interference task was used for two very different purposes in study 1 and 2: in study 1 it was used as part of a reconsolidation paradigm to modify existing memories; in study 2 the same protocol was used not to modify memory, but rather to enhance the contrast between conditions in the subsequent test. If this is correct, it is worth mentioning, because these very different purposes (to edit memory traces in study 1 and to enhance signal to noise in study 2) are somewhat confusing.

Yes, that is correct. Thank you for pointing this out. We have now added a sentence explaining this difference as follows (lines 810-814):

“Thirty minutes after the interference task was finished, participants completed the testing session, in which their memory of the original task was assessed. Please note that the interference task in Study 2 was not used to impair memory stabilization as in Study 1, but rather to enhance the contrast between reminder conditions in the *subsequent test*.”

19. Line 639 - it may be justified to cite a recent paper suggesting this very model (Lewis and Bendor, Curr Bio, 2019).

Thank you for pointing this out. We have added a sentence describing the Lewis and Bendor model as follows (lines 637-641):

“This idea is in keeping with an elegant model suggested by Lewis and Bendor. This model proposes that external sensory cues are processed in the neocortex during sleep, which effectively tricks the hippocampus by evoking what looks like a spontaneous reactivation of a memory in the cortex, in turn influencing hippocampal replay⁵⁶.”

20. The authors relate their findings to those of Schreiner & colleagues (2015) and explain the lack of a cueing effect in that paper as further evidence for partial cueing being more beneficial. However, that study also showed that presenting a tone after the partial cue inhibits consolidation. It seems as though it is not the partial aspect of the cue, but rather the interference between the different parts of the cue, that best explain the lack of a TMR effect. Therefore, it is less likely that the results reflect a partial vs full cueing difference like the one in the current manuscript.

This is an interesting point. We believe that it may be a combination of both processes that determines the reactivation effect, i.e. the partial nature of the cue and the silent refractory period after the partial cue. The TMR effect may not be observed when either a complete cue is presented because this doesn't even labilize the memory or the partial cue with a subsequent sound is presented because the sound disrupts the processing of the partial cue. However, in order to test this hypothesis, a new series of experiments should be designed (with shorter sounds) varying the type of reminder and interrupting or not the refractory period. We added an alternative explanation of the results in the discussion section (lines 694-708):

“Alternatively, it could be argued that in the present study, it is not the incomplete aspect of the reminder, but rather the interference between different parts of the complete reminder, that best explains the lack of a cueing effect for complete reminders. This interpretation is based on an additional finding by Schreiner and colleagues⁴⁷, showing that the presentation of a tone after the single cue likewise inhibits consolidation. We suggest that it may be a combination of both processes that determines the reactivation effect, i.e. the incomplete aspect of the cue and a silent refractory period after the incomplete cue. The cueing effect may not be observed when either a complete cue is presented, because complete cues do not induce a mismatch, or an incomplete cue is presented with a subsequent sound, because then the sound disrupts the processing of the incomplete cue. Future studies should vary the type of reminder with or without an interruption of the refractory period and examine the exact time frame and the contribution of different sleep stages for memory reactivation during sleep.”

21. The authors mention that MMN still persists (partly) during sleep and that

this supports their hypothesis that prediction errors play a part in consolidation. However, MMN has mostly been observed in light stages of sleep and in REM sleep (e.g., Scultholpe et al., 2009). Therefore, this evidence does not support the suggested model, which concentrates on consolidation during NREM.

Yes indeed, almost all papers showing MMN during sleep focused on light stages of sleep. However, there are other papers showing the same effect during SWS. We have changed the discussion section as follows (lines 591-604):

“While the P300 and the intermediate mismatch peak of the MMN were found to be abolished during sleep, the early and late parts of the MMN are preserved during sleep⁴⁵. Although most studies focused on mismatch detection during light sleep stages^{45,62}, some studies observed signs of mismatch detection also during SWS⁶³⁻⁶⁵, suggesting that at least a rudimentary form of mismatch detection may still be in place during SWS. Unfortunately, the present experimental design did not allow for a direct analysis of mismatch detection during sleep with event-related potentials analyses because of a low number of reactivation trials, varying lengths between cues (syllables/words for incomplete/complete reminders) and the number of rejected trials after data preprocessing. However, we showed that both the incomplete and complete reminder cues evoked consistent increases in theta band power, followed by responses in the spindle band. These results are in line with prior reports from auditory cueing studies^{47,48,66,67}.”

As a side note, the citation in line 654 appears to be wrong.

Thank you, we have corrected this mistake.

Reviewer #2.

Minor:

1) The experimental paradigm drawing in fig 3 A is confusing. For instance, it is not clear that the forward stripy box is sleep while backwards stripey is wake. It really took me awhile to figure this out. Please rework this figure, label everything better, and try to make it clearer.

We agree and have reworked the figure accordingly. We hope the drawing is clearer now.

“Figure 3. Memory stabilization by different reminder cues during sleep in Study 2. (a) In three experiments, all participants were trained in the evening before a 40-min sleep period during which the complete reminder (Rc groups), incomplete reminder (Ri groups) or no reminder (NR groups) was presented. In Exp. 2, participants were awakened after the 40-min sleep period, learned the interference task 30 min later and were tested on the original task another 30 min later. In Exp. 3, procedures were exactly the same for training and reminder presentation, but participants were allowed to continue sleeping after reminder presentation and were awakened in the morning, with interference learning and testing taking place thereafter. In Exp. 4, procedures were also identical for training and reminder presentation, but participants were awakened after the 40 min sleep period and remained awake for another 7 hours, with interference learning and testing taking place thereafter. **(b)** In Exp. 2 (40min), both reminder groups performed better than the no reminder group, with no difference between the complete and incomplete reminder groups (left panel). In Exp. 3 (8h), the incomplete reminder group performed better than the complete reminder and no reminder groups, with no difference between the complete and no reminder groups (middle panel). In Exp. 4 (40min-7h), the incomplete reminder group performed better than the complete reminder group and showed a trend toward better performance than the no reminder group, with no difference between the complete and no reminder groups (right panel). Memory change represents the number of correct responses at testing minus training, i.e. indicating forgetting. Means ± SEM are shown. * $P < 0.05$; # $P = 0.072$.”

2) When describing the exploratory analysis please give more detail about the factor in our ANOVA (was it all 3 experiments)?

Yes, it was indeed all 3 experiments. We have added the number of factors for the exploratory analysis (lines 1024-1028):

“For additional exploratory cross-experiment analysis, a two-way ANOVA with the between-subjects factors “reminder type” (complete reminder, incomplete reminder, no reminder) and “experiment” (experiment 2, 3, 4) was conducted, followed by post-hoc tests. Analyses were performed with SPSS (version 22.0). Alpha was set to 0.05.”

3) Figure 4: it would be good to see a plot of the comparison between complete and incomplete – e.g. the spindles were significantly different and it would be nice to see this represented.

We thank the reviewer for this suggestion. We have now included a plot of the comparison between complete and incomplete reminder conditions and we have modified the manuscript accordingly (lines 428-434):

“To test for differences in brain oscillation patterns upon reactivation with the different reminder types, time-frequency analyses were performed for each condition: Rc (complete reminder), Ri (incomplete reminder), and NR (no reminder). EEG data for the reactivation period were contrasted using equivalent time windows between the NR and Rc conditions (Fig. 4a), and between the NR and Ri conditions (Fig. 4b), as well as between the Rc and Ri conditions (Fig. 4c).”

Figure 4. Reminder cues during SWS evoke responses in the theta and spindle band. EEG power response over central sites aligned to cue onset ($t = 0$ sec, representing onset of the entire word for the complete reminder and the first syllable for the incomplete reminder, respectively). Note that reminder conditions are identical during sound presentation and only differ from the start of cue presentation. Grand

averages from all experiments of Study 2 are shown for (a) the complete reminder conditions and (b) the incomplete reminder conditions. Colors show power changes relative to pre-stimulation baseline between -4 and -3 s. Black waveform shows evoked time-domain response. Data are masked by cluster permutation statistics contrasting stimulation time windows to comparable time windows without stimulation in the no reminder conditions. Cluster-level statistics (sample-level threshold of $P = 0.01$) for a: sound-evoked spindle cluster $P = 0.032$, cue-evoked spindle and theta clusters $P < 0.001$; for b: sound-evoked spindle cluster $P = 0.029$, cue-evoked spindle and theta clusters $P < 0.001$. (c) Comparison between data from both conditions (Incomplete reminder subtracted from and statistically compared to Complete reminder). Cluster-level statistics (using a sample-level threshold of $P = 0.01$) for cue-evoked spindle cluster: $P = 0.025$. Musical notes represent onset of the sound, loudspeakers represent onset of the cue (word/syllable).

4) Were there any correlations between spindle/theta power and TMR effects? This would be particularly interesting given the difference in response to complete/incomplete cues.

Thank you for this suggestion. We have performed the correlation analysis between the observed spindle and theta power increases in the time-frequency analysis and memory change as suggested. However, we did not observe any significant correlations, except for a correlation of memory change with cue-evoked theta power in the complete reminder-40min/7h group. We have included this finding in the revised manuscript as follows (lines 498-505):

“We further analyzed correlations between memory change and the observed spindle and theta power increases upon reactivation with the different reminder cues (as reported in Fig. 4). This analysis revealed no significant correlations, except for an association of memory change with the cue-evoked theta power increase in the “complete reminder-40min/7h” group ($r = -0.64$, $P = 0.047$). However, it should be noted that none of the correlations were corrected for multiple comparisons and should therefore be interpreted with caution.”

5) The point about wake being equivalent to REM is important and could have more air time. Were there correlations between behavioural findings and sleep stage minutes or percentages?

We thank the reviewer for this helpful comment. In the revised version of the manuscript, we have included correlations between time spent in the different sleep stages, power density in the frequency bands of interests (slow oscillation, delta and

spindles) and memory change. We incorporated this information in the materials and methods and result section and included an alternative explanation of the results.

Lines 487-505: “Correlation analyses

To test whether the observed behavioral effects in memory performance are associated with changes in sleep measures, we performed correlation analyses between memory change and time spent in single sleep stages as well as power in *specific frequency bands during the entire sleep period*. In the “no reminder-8h” group, higher scores in memory change were associated with higher percentage of SWS (Fig. 5a, $r = 0.67$, $P = 0.008$), as well as with higher slow oscillation power (0.5-1 Hz; Fig. 5b, $r = 0.55$, $P = 0.03$), higher delta power (1-4 Hz; Fig. 5c, $r = 0.56$, $P = 0.0029$), and higher spindle power density (9-15 Hz; Fig. 5d, $r = 0.64$, $P = 0.01$). In the “no reminder-40 min” group, memory change was only associated with higher spindle power density (9-15 Hz; Fig. 5e, $r = 0.64$, $P = 0.035$). There were no other significant correlations. We further analyzed correlations between memory change and the observed spindle and theta power increases upon reactivation with the different reminder cues (as reported in Fig. 4). This analysis revealed no significant correlations, except for an association of memory change with the cue-evoked *theta power increase in the “complete reminder-40min/7h” group* ($r = -0.64$, $P = 0.047$). However, it should be noted that none of the correlations were corrected for multiple comparisons and should therefore be interpreted with caution.

Figure 5. Sleep and memory performance. *Significant correlations in the “no reminder/8h” group between memory change and (a) percentage of SWS, (b) slow oscillation power in NREM, (c) delta power in NREM, and (d) spindle power in NREM. (e) Significant correlation in the “no reminder/40min” group between memory change and spindle power in NREM.*

Lines 618-637: “Although we did not find any consistent associations between the spindle and theta responses upon reminder presentation and memory performance, we did observe associations between memory performance and other sleep measures. Better memory performance was associated with higher amounts of SWS as well as *with higher power in the slow oscillation, delta and spindle band in the “no reminder-8h” group. In the “no reminder-40min” group, better memory performance was associated with power in the spindle band.* Interestingly, correlations with memory performance were only observed in the groups that received no reactivation. Indeed, it is well known that the time spent in SWS as well as the power of slow oscillations, delta and spindles is related to sleep-dependent memory improvement⁵⁷. It was also found that declarative memory consolidation particularly depends on the duration of SWS and that external reactivation cues can accelerate the consolidation process⁵⁸. It could be speculated that without externally triggered reactivation, the amount of SWS as well as associated slow oscillations, delta and spindle activity, makes a difference for memory consolidation because of more or less spontaneous reactivation events. Externally triggered reactivation, on the other hand, may speed up this process and facilitate *stronger “artificial” reactivations such that the improvement of performance does not depend on spontaneous reactivations so much anymore.*”

Lines 982-998: “Correlations with sleep stages and power density

We further performed Pearson correlations between percentage in different sleep stages and memory change for all the groups of Study 2. Moreover, we performed Pearson correlations between memory change and power density in the frequency bands of interest (slow oscillation: 0.5-1 Hz; delta: 1-4 Hz; theta: 4-8 Hz; spindles: 9-15 Hz) during NREM sleep for all groups of Study 2 .

For calculating individual average power spectra, the data was analyzed using SpiSOP (<https://www.spisop.org>, RRID:SCR_015673) that is based on code of FieldTrip⁷⁹ (<http://fieldtriptoolbox.org>, RRID:SCR_004849) in MATLAB 2013b (Mathworks, Natick, USA). Artifact-free NREM epochs were divided into consecutive 10 s blocks that overlapped 5 s in time. Each block was tapered by a single Hanning window before applying Fast Fourier Transformation that resulted in block power spectra with a frequency resolution of 0.1 Hz. Power spectra were then averaged *across all blocks (Welch’s method)*. *Mean power over central electrodes was determined for the frequency bands of interest, slow oscillations (0.5–1 Hz); delta (1-4 Hz), theta (4-8 Hz) and spindle range (9–15 Hz).*”

Lines 1029-1032: “In order to examine if the time spent in the different sleep stages and power of the frequencies of interest were related to memory change, we performed

bilateral Pearson correlations. The correlations were not corrected for multiple *comparisons*.”

Considering that we did not observe any correlations with REM sleep and that Reviewer 1 pointed us to an alternative explanation (comment #D), we decided not to go deeper into our REM sleep hypothesis. We have added the following in the discussion section (lines 659-667):

“Our data extend this view in proposing that late periods of wakefulness may be similarly effective than REM sleep in supporting memory stabilization for the long-term. However, assuming that the memory benefit in the incomplete reminder condition is established already after 40 minutes of sleep and remains unchanged thereafter, the difference between reminder types could be alternatively explained by a memory decay for the complete reminder condition during the extended retention interval. These two alternative explanations cannot be discerned with the present experimental design and should be subject to further investigation.”

Reviewer #3:

Major comments:

Comment 1: As far as I understood, the authors propose that the underlying process driving their results (thus the effectiveness of incomplete reminders to labilize memories during wake and stabilize memories during sleep) is a prediction error / mismatch detection mechanism.

To induce such mismatch perception during wakefulness, participants of study 1 attended on the day after the learning session a reactivation session. Here they received reminders of the learning task (either complete reminder or incomplete reminder). For the incomplete reminder condition only the first syllable of each word which were associated before with a sound was presented. This procedure was meant to induce a prediction error and consequently reconsolidation in the participants, assuming that they would have expected to hear the whole word instead of a cropped version.

Unfortunately, I have some struggles to follow this reasoning. As far as I can see, a very similar procedure was already administered during memory retrieval the day before. Here, participants were presented with sounds plus the first syllable of the associated words (thus a incomplete reminder) and had to vocalize the missing part of each word. Thus, participants were already quite familiar with being presented with incomplete reminders, rendering further expectation violations rather unlikely.

We apologize for not having made this clearer. In fact, the reactivation procedure was not identical to the recall procedure. Like the reviewer correctly pointed out, during recall, subjects had to vocalize the missing part of each word, which is what they also predicted to do in the reactivation session. In the reactivation session in Study 1, subjects were asked to respond to the first syllable as during recall, i.e. they were instructed to vocalize the entire word. However, other than during the recall session, each trial was interrupted and subjects never actually had the opportunity to respond, even though they expected to respond. This discrepancy should induce a prediction error as has been shown in previous reconsolidation studies from our group (Forcato et al. 2016; Forcato et al., 2014). That said, we agree that the reasoning is a bit trickier for the sleep experiment, because subjects did not receive any instructions for the reactivation during sleep.

We have clarified the difference between recall and reactivation in the manuscript as follows (lines 892-902):

“It is important to highlight that the reactivation session differed from the cued recall procedure during the training session in two regards: 1) participants expected to respond as in the cued recall procedure but the microphone indicating the opportunity to respond never appeared on the screen and 2) each trial was interrupted without allowing the participants to respond, a procedure known to induce a prediction error, which triggers memory labilization/reconsolidation^{8,15}. After the reactivation session, almost all participants reported to the experimenter that the microphone did not appear on the screen and that they could never respond, some of them even thought that there was a technical problem, suggesting that subjects actually expected to be required to *respond until the end of the reactivation session.*”

Apart from that even if they were not familiar with the procedure, it seems likely that presenting incomplete reminders could lead to a mismatch during the first couple of **trials. But could one really assume that there is still such a prediction error after let's say 20 trials?** I would be quite interested about the thoughts of the authors concerning this topic.

This is an interesting question. At some point, subjects probably realized that they would never have to respond. However, each trial started new with the same instruction and they would never know if the image of the microphone would appear on the screen and they would have to respond at some point. Moreover, almost all participants in the first study reported to the experimenter that the microphone did not

appear on the screen and that they could never respond, some of them were even worried and thought that there was a technical problem. Thus, according to the report of the subjects after the reactivation session we believe that they actually expected the microphone to appear until the end of the reactivation session. We have addressed this issue in the manuscript as follows (lines 898-902):

“After the reactivation session, almost all participants reported to the experimenter that the microphone did not appear on the screen and that they could never respond, some of them even thought that there was a technical problem, suggesting that subjects actually expected to be required to respond until the end of the reactivation session.”

Comment 2: My second concern is quite related to the one above. The fact that only incomplete reminders presented during sleep showed consistent effects on memory stabilisation led the authors to suggest that also here some sort of mismatch detections should be active during sleep, with the prediction error tagging memories for late consolidation. While it is known that prediction errors might be elicited at least to some extent during sleep (manifesting in distinct ERPs as discussed by the authors), I just don't see why the memory system should be geared towards mismatch detection during sleep. Or in other words the authors present no data in clear favour of this explanation.

Thus, if the authors believe that a mismatch detection process is driving their results, I wonder why they did not check in their sleep data whether they can find any difference between the ERPs related to complete and incomplete reminders. This would tremendously help their argumentation.

We thank the reviewer for this comment and we agree that our interpretation would be strengthened with supporting data from ERP analyses. Unfortunately, the present experimental design did not allow for a direct analysis of mismatch detection during sleep with ERP analyses because of methodological limitations. First of all, we have very few reactivation trials to perform a proper ERP analysis (30 per participant), some trials even had to be excluded during preprocessing due to artefacts. Moreover, the syllables and the words in the incomplete and complete reminder conditions, respectively, substantially differ in length and meaning, with the entire words constituting longer and more semantically meaningful cues. Thus, any differences in the ERP between incomplete and complete reminder conditions could be due to a number of different factors apart from mismatch detection. Future studies should test this question more systematically, using experimental paradigms that are optimized for ERP analyses. Nevertheless, we believe that our data can arguably still be interpreted in line with the mismatch detection theory, considering that some studies have shown

mismatch detection even during SWS. That said, we concur that future studies may show that our interpretation is wrong. We have addressed this issue in the manuscript as follows (lines 591-603):

“While the P300 and the intermediate mismatch peak of the MMN were found to be abolished during sleep, the early and late parts of the MMN are preserved during sleep⁴⁵. Although most studies focused on mismatch detection during light sleep stages^{45,62}, some studies observed signs of mismatch detection also during SWS⁶³⁻⁶⁵, suggesting that at least a rudimentary form of mismatch detection may still be in place during SWS. Unfortunately, the present experimental design did not allow for a direct analysis of mismatch detection during sleep with event-related potentials analyses because of a low number of reactivation trials, varying lengths between cues (syllables/words for incomplete/complete reminders) and the number of rejected trials after data preprocessing. However, we showed that both the incomplete and complete reminder cues evoked consistent increases in theta band power, followed by *responses in the spindle band. These results are in line with prior reports...*”

Comment 3: I have some concerns with regards to the EEG analyses and their interpretation.

-The authors report that EEG segments containing the reminder cues were contrasted against comparable time-windows of the no-reminder control group. They do not give any more detail how that was achieved, even though it would help to understand the shown figure.

We apologize for not having made this clearer. In the no reminder control group, markers in the EEG were introduced at positions comparable to the markers for reminder presentation in the reminder groups, i.e. at the same time points relative to sleep onset. To perform the statistical comparisons, for each participant, an average power for each time point and frequency was estimated; each time-frequency sample was then compared between groups. We included a clearer explanation about how the figure was done.

In the results section, it now reads as follows (lines 427-455):

“Electrophysiological analysis of cued reactivation during sleep

To test for differences in brain oscillation patterns upon reactivation with the different reminder types, time-frequency analyses were performed for each condition: Rc (complete reminder), Ri (incomplete reminder), and NR (no reminder). EEG data for

the reactivation period were contrasted using equivalent time windows between the NR and Rc conditions (Fig. 4a), between the NR and Ri conditions (Fig. 4b), as well as between the Rc and Ri conditions (Fig. 4c). Following the presentation of the cue, both complete and incomplete reminder cues (i.e. words and syllables, respectively) evoked responses in the theta band (4-8 Hz) and the fast spindle band (12-16 Hz, all cluster-level $P < 0.001$) as compared to the no reminder condition. When both reminder conditions were directly compared, cue-evoked power increases in the slow spindle band (10-12 Hz) were stronger in the complete reminder compared to the incomplete reminder condition (Fig. 4c, $P = 0.025$), whereas power increases in the theta band (4-8 Hz) did not differ significantly ($P = 0.15$). Regarding responses evoked by the presentation of the sound (which was identical for both reminder conditions), significant power increases were detected in the fast spindle band in both reminder conditions (12-16 Hz, Rc vs. NR: Fig. 4a $P = 0.032$, Ri vs. NR: Fig. 4b $P = 0.029$); yet despite descriptive increases in theta power in response to the sound, there were no significant clusters spanning that frequency range (Rc vs. NR: $P = 0.40$, Ri vs. NR: $P = 0.95$). Interindividual differences and variability in the sound recordings may have introduced variance obscuring sound-induced power changes, which have typically been observed in previous studies using sounds as reactivation cues⁴⁶. However, a genuinely attenuated theta response would also be in line with previous research, suggesting that additional sounds immediately following a reminder cue can disrupt processing of that initial reminder⁴⁷⁻⁴⁸. Long-lasting auditory stimulation like the sounds used in the present study may yield a similar effect.”

Figure 4. Reminder cues during SWS evoke responses in the theta and spindle band. EEG power response over central sites aligned to cue onset (t = 0 sec,

representing onset of the entire word for the complete reminder and the first syllable for the incomplete reminder, respectively). Note that reminder conditions are identical during sound presentation and only differ from the start of cue presentation. Grand averages from all experiments of Study 2 are shown for (a) the complete reminder conditions and (b) the incomplete reminder conditions. Colors show power changes relative to pre-stimulation baseline between -4 and -3 s. Black waveform shows evoked time-domain response. Data are masked by cluster permutation statistics contrasting stimulation time windows to comparable time windows without stimulation in the no reminder conditions. Cluster-level statistics (sample-level threshold of $P = 0.01$) for a: sound-evoked spindle cluster $P = 0.032$, cue-evoked spindle and theta clusters $P < 0.001$; for b: sound-evoked spindle cluster $P = 0.029$, cue-evoked spindle and theta clusters $P < 0.001$. (c) Comparison between data from both conditions (Incomplete reminder subtracted from and statistically compared to Complete reminder). Cluster-level statistics (using a sample-level threshold of $P = 0.01$) for cue-evoked spindle cluster: $P = 0.025$. Musical notes represent onset of the sound, loudspeakers represent onset of the cue (word/syllable).

In the methods section, we have clarified as follows (lines 957-980):

“Reactivation EEG data processing

All experiments in Study 2 diverged only after the reactivation phase. Therefore, electrophysiological data for the reactivation period of the three experiments were pooled for each condition, resulting in three datasets: complete reminder (Rc), incomplete reminder (Ri) and no reminder (NR). EEG data of each reactivation event was cut into 12-sec trials (-7 to 5 sec, with $t = 0$ sec referring to cue onset). Because the reminder conditions are identical during the sound presentation interval and only differ upon presentation of the cue (syllable/word), the analyses were time-locked to cue onset. Trials containing artefacts in any of the channels were removed using automatic and visual rejection. After data preprocessing, EEG data from 108 subjects remained for further analyses, overall 16 reactivation trials were excluded for Rc, 10 trials for Ri and 12 trials for NR. Time-frequency analysis was performed (10-cycle Morlet wavelets) to calculate the power for the average of the electrodes C3 and C4 relative to a baseline of one second right before sound onset (-4 and -3 sec). This processing resulted in the relative power of each frequency at each time point for Rc, Ri and NR independently. All reactivations were aligned to cue onset ($t=0$ sec). Significant responses upon the sound and cue presentation in the Rc and Ri conditions were compared to the NR condition as well as between both reminder conditions. *Responses upon the sound presentation are called “sound-evoked” responses and responses upon the cue presentation are called “cue-evoked” responses. Different*

significant clusters were detected, according to the classical band frequencies for slow spindles (10-12 Hz), fast spindles (12-16 Hz) and theta (4-8 Hz).”

Lines 1033-1040: “To assess the statistical differences between the EEG response to the reactivation in each condition, the time-frequency values previously calculated (see Reactivation EEG processing) were compared using sample-level two-tailed independent-samples t-tests followed by a non-parametric cluster-permutation procedure to correct for multiple comparisons⁸⁰ (5000 permutations), as implemented in the open-source toolbox FieldTrip⁷⁹. All EEG data processing was done using Matlab 2017a (MathWorks Inc, Natick, Massachusetts, USA).”

- The authors report that each reminder (30 in total) was presented once, leading to a rather low stimulus number. Given that trials were rejected during pre-processing, it would be good to know how many of them entered the final analysis.

Thank you. We have included a paragraph in Materials and Methods with the number of reactivation trials after artifact rejection and we now also discuss this limitation of the EEG analysis (lines 965-969):

“Trials containing artefacts in any of the channels were removed using automatic and visual rejection. After data preprocessing, EEG data from 108 subjects remained for further analyses, overall 16 reactivation trials were excluded for Rc, 10 trials for Ri and 12 trials for NR.”

Lines 596-603: “Unfortunately, the present experimental design did not allow for a direct analysis of mismatch detection during sleep with event-related potentials analyses because of a low number of reactivation trials, varying lengths between cues (syllables/words for incomplete/complete reminders) and the number of rejected trials after data preprocessing. However, we showed that both the incomplete and complete reminder cues evoked consistent increases in theta band power, followed by *responses in the spindle band. These results are in line with prior reports...*”

- In my view the biggest problem concerning the EEG results is that it is entirely unclear whether they comprise any memory-relatedness. The authors did not present any memory unrelated control stimuli during sleep, which could have served as meaningful contrast / control condition. And I assume that the low trial number hinders them from contrasting their stimuli according to a subsequent memory rationale. Thus, I fear that the main activation which can be seen in Figure 4 is in main parts related to the processing of auditory stimuli during sleep.

We agree with the reviewer that this is a major limitation of our experimental design. Indeed, the number of trials is too low for an analysis of the subsequent memory effect. Nevertheless, we have performed a correlation analysis between the observed spindle and theta power increases in the time-frequency analysis and memory change as suggested. We did not observe any significant correlations, except for a correlation of memory change with cue-evoked theta power increase in the “complete reminder-40min/7h” group. However, we have also run a correlation analysis between memory change and time spent in the different sleep stages as well as power density in the frequency bands of interests (slow oscillation, delta and spindles) during the entire sleep period. This analysis revealed associations between memory performance and SWS, slow oscillations, delta and spindle power in the no reminder groups, which we have now reported in the revised manuscript. Furthermore, we have discussed this issue in more detail in the discussion section and have also provided an alternative explanation of our results:

In the results section, it now reads as follows (lines 498-505):

“We further analyzed correlations between memory change and the observed spindle and theta power increases upon reactivation with the different reminder cues (as reported in Fig. 4). This analysis revealed no significant correlations, except for an association of memory change with the cue-evoked theta power increase in the “complete reminder-40min/7h” group ($r = -0.64$, $P = 0.047$). However, it should be noted that none of the correlations were corrected for multiple comparisons and should therefore be interpreted with caution.”

Lines 487-498: **“Correlation analyses**

To test whether the observed behavioral effects in memory performance are associated with changes in sleep measures, we performed correlation analyses between memory change and time spent in single sleep stages as well as power in specific frequency bands during the entire sleep period. In the “no reminder-8h” group, higher scores in memory change were associated with higher percentage of SWS (Fig. 5a, $r = 0.67$, $P = 0.008$), as well as with higher slow oscillation power (0.5-1 Hz; Fig. 5b, $r = 0.55$, $P = 0.03$), higher delta power (1-4 Hz; Fig. 5c, $r = 0.56$, $P = 0.0029$), and higher spindle power density (9-15 Hz; Fig. 5d, $r = 0.64$, $P = 0.01$). In the “no reminder-40 min” group, memory change was only associated with higher spindle power density (9-15 Hz; Fig. 5e, $r = 0.64$, $P = 0.035$). There were no other significant correlations.”

Figure 5. Sleep and memory performance. *Significant correlations in the “no reminder/8h” group between memory change and (a) percentage of SWS, (b) slow oscillation power in NREM, (c) delta power in NREM, and (d) spindle power in NREM. (e) Significant correlation in the “no reminder/40min” group between memory change and spindle power in NREM.*

In the discussion section, it now reads as follows (lines 618-637):

“Although we did not find any consistent associations between the spindle and theta responses upon reminder presentation and memory performance, we did observe associations between memory performance and other sleep measures. Better memory performance was associated with higher amounts of SWS as well as with higher power in the slow oscillation, delta and spindle band in the “no reminder-8h” group. In the “no reminder-40min” group, better memory performance was associated with power in the spindle band. Interestingly, correlations with memory performance were only observed in the groups that received no reactivation. Indeed, it is well known that the time spent in SWS as well as the power of slow oscillations, delta and spindles is related to sleep-dependent memory improvement⁵⁷. It was also found that declarative memory consolidation particularly depends on the duration of SWS and that external reactivation cues can accelerate the consolidation process⁵⁸. It could be speculated that without externally triggered reactivation, the amount of SWS as well as associated slow oscillations, delta and spindle activity, makes a difference for memory consolidation because of more or less spontaneous reactivation events. Externally triggered reactivation, on the other hand, may speed up this process and facilitate stronger “artificial” reactivations such that the improvement of performance does not depend on spontaneous reactivations so much anymore.”

In the methods section, it now reads as follows (lines 982-998):

“Correlations with sleep stages and power density

We further performed Pearson correlations between percentage in different sleep stages and memory change for all the groups of Study 2. Moreover, we performed Pearson correlations between memory change and power density in the frequency bands of interest (slow oscillation: 0.5-1 Hz; delta: 1-4 Hz; theta: 4-8 Hz; spindles: 9-15 Hz) during NREM sleep for all groups of Study 2 .

For calculating individual average power spectra, the data was analyzed using SpiSOP (<https://www.spisop.org>, RRID:SCR_015673) that is based on code of FieldTrip⁷⁹ (<http://fieldtriptoolbox.org>, RRID:SCR_004849) in MATLAB 2013b (Mathworks, Natick, USA). Artifact-free NREM epochs were divided into consecutive 10 s blocks that overlapped 5 s in time. Each block was tapered by a single Hanning window before applying Fast Fourier Transformation that resulted in block power spectra with a frequency resolution of 0.1 Hz. Power spectra were then averaged *across all blocks (Welch's method)*. *Mean power over central electrodes was determined for the frequency bands of interest, slow oscillations (0.5–1 Hz); delta (1-4 Hz), theta (4-8 Hz) and spindle range (9–15 Hz).*”

Lines 1029-1032: “In order to examine if the time spent in the different sleep stages and power of the frequencies of interest were related to memory change, we performed bilateral Pearson correlations. The correlations were not corrected for multiple comparisons.”

The result that complete reminders tended to elicit stronger activity in the sleep spindle range might just be related to the fact that those stimuli were longer in duration as compared to incomplete reminders. This would also fit to the known sleep-preserving role of sleep spindles (e.g. Dang –Vu et al., 2010). Thus stimuli of longer duration might just have a generally higher capability to wake participants up and thus elicit preferentially sleep spindles, which in turn protect the sleeping brain.

We thank the reviewer for this interesting suggestion and agree that this might be an alternative explanation. We have now addressed this issue in the discussion section as follows (lines 612-617):

“However, it is also known that sleep spindles play a role in preserving sleep⁶⁸. Since complete reminders were of slightly longer duration, these reminders might have a generally higher tendency of inducing arousals and thus elicit stronger spindle responses to prevent subjects from waking up. Thus, the difference in responses between incomplete and complete reminder cues should be interpreted with caution.”

- I was wondering why the EEG analysis was time-locked to the word cues and not the **preceding sounds. Wouldn't the authors assume that hippocampal pattern completion** and hence reactivation processes should already (and preferentially) be elicited by these initial cues?

This is an important point. We agree that pattern completion processes may already be initiated upon the presentation of the sound. However, the complete and incomplete reminder conditions are entirely identical during the sound presentation interval. The difference between conditions by design only starts with the presentation of the cue (syllable/word). Therefore, we would not expect any differences between conditions during the sound presentation interval and this is also the reason why we ran the analyses time-locked to the cue. We have included a sentence in the Materials and Methods section explaining this issue and also highlighted this fact in the legend of Figure 4.

Methods section (lines 961-965):

"EEG data of each reactivation event was cut into 12-sec trials (-7 to 5 sec, with $t = 0$ sec referring to cue onset). Because the reminder conditions are identical during the sound presentation interval and only differ upon presentation of the cue (syllable/word), the analyses were *time-locked to cue onset*."

Figure 4 legend (lines 467-485):

"Figure 4. Reminder cues during SWS evoke responses in the theta and spindle band. EEG power response over central sites aligned to cue onset ($t = 0$ sec, representing onset of the entire word for the complete reminder and the first syllable for the incomplete reminder, respectively). Note that reminder conditions are identical during sound presentation and only differ from the start of cue presentation. Grand averages from all experiments of Study 2 are shown for (a) the complete reminder conditions and (b) the incomplete reminder conditions. Colors show power changes relative to pre-stimulation baseline between -4 and -3 s. Black waveform shows evoked time-domain response. Data are masked by cluster permutation statistics contrasting stimulation time windows to comparable time windows without stimulation in the no reminder conditions. Cluster-level statistics (sample-level threshold of $P = 0.01$) for a: sound-evoked spindle cluster $P = 0.032$, cue-evoked spindle and theta clusters $P < 0.001$; for b: sound-evoked spindle cluster $P = 0.029$, cue-evoked spindle and theta clusters $P < 0.001$. (c) Comparison between data from both conditions (Incomplete reminder subtracted from and statistically compared to Complete reminder). Cluster-level statistics (using a sample-level threshold of $P = 0.01$) for cue-evoked spindle

cluster: $P = 0.025$. Musical notes represent onset of the sound, loudspeakers represent *onset of the cue (word/syllable)*.”

Lines 470-472: *“Note that reminder conditions are identical during sound presentation and only differ from the start of cue presentation.”*

In sum I don't see that the EEG analyses and results in the present form help to understand the behavioural effects reported by the authors by any means.

In this new version of the manuscript we have included correlation analyses between time in SWS, power density in frequencies of interest (SO, delta, spindle) and memory change, as we reported above. We also performed correlation analyses between power increases during reactivation and memory change as suggested. We believe that these new analyses have substantially improved the manuscript. However, we agree that it is important to highlight the limitations of our paradigm. To mention only a few, in the present study, the reminders are rather long and may obscure the refractory period after initial reminder onset; the cues (syllables vs. words) differ in length and semantic meaning, making it difficult to compare ERPs between conditions; the number of reactivation trials is too low for optimal EEG/ERP analysis. All of these limitations are now addressed in the revised manuscript (as outlined above) and we hope that the exact mechanisms underlying the observed effects will be subject to future investigation. We are convinced that the results presented in the revised version of the manuscript provide a new perspective into the study of boundary conditions to improve memory during the TMR procedure, which will hopefully stimulate further research.

Minor comment: The authors state in the introduction that ...the mechanisms underlying memory stabilization upon cued reactivation in the sleep state are largely unknown... Maybe the authors would want to include the work of Bendor& Wilson (2012) as well as recent findings by Rothschild and colleagues (2017), which sheds some light on the neural processes associated with targeted memory reactivation.

We thank the reviewer for this suggestion. We have added these works in the introduction section as follows (lines 82-88):

“Despite compelling evidence for the beneficial effects of cued memory reactivation during sleep, there are only few studies on the mechanisms underlying memory stabilization upon cued reactivation in the sleep state. These studies show that hippocampal replay is linked to cued reactivation²⁷ and that cortical input during

reactivation is part of a cortico–hippocampal–cortical loop, strengthening memory traces through the reverberation of replay between the cortex and hippocampus³⁹.”

REVIEWERS' COMMENTS:

Reviewer #1 (Remarks to the Author):

The authors appropriately responded to all the changes and incorporated important changes to the manuscript. This has improved the manuscript significantly, but there are still several issues that should be addressed before this manuscript should be accepted.

1. Readers may be confused by the use of the terms "study" and "experiment" in this manuscript. For example, in line 247, the reader may assume that experiment 2 and study 2 are the same, and they may be confused when experiment 3 is introduced in line 343. I suggest adding a short paragraph at the beginning of the Results section for study 2, explaining it is composed of experiments 2-4 that will be described in order.

2. The authors clarified what they mean by "a 40-minute nap", but this interpretation is non-trivial and may be misleading to authors who don't carefully read the results section. In some cases, the phrasing is simply erroneous (e.g., line 380 states that there were 40 minutes of sleep, when in fact for some participants there were more than twice as many). I suggest one of the following: either define a term, such as "short-duration sleep", which will be introduced along with a short explanation and then used consistently; or rephrase as "at least 40 minutes" in key positions in the manuscript, along with a referral to the methods section. If the authors are set on keeping the current phrasing, I suggest adding a note at the beginning of the Results for study 2 explaining that 40-minute sleep periods are not limited to 40 minutes.

3. In my previous point #8, I noted that there is some contradiction between the exclusion justification in line 751 and the fact that arousals were immediately followed by a temporary termination of cuing. This issue has not been resolved. Waking up involves an arousal, followed by a break in cuing. Assumingly, if the participants went back to sleep, cuing should start again and the participants should be included in the final sample. Perhaps the authors mean that some participants were not sufficiently cued because they did not go back into N3 within some timeframe. If so, they should mention what that timeframe was (in the current methods section, there is no upper boundary for the length of a sleep period) and rephrase the omission criteria. Alternatively, the authors may have excluded any participant that had even a single epoch of wake after sleep onset. This should be clarified and justified.

4. For the time-frequency analysis, equivalent NR time epochs were used as baseline. The authors should mention how those were chosen (e.g., were they chosen from N3? Were epochs within the first 10 minutes of stable SWS chosen or excluded like for the cued data? Were these epochs non-overlapping?)

Reviewer #2 (Remarks to the Author):

Thanks for this good revision. You have answered all of my concerns, and I am now satisfied that the paper should be published.

Reviewer #3 (Remarks to the Author):

The authors addressed all my concerns. I have nothing more to add.

Response to Reviewers

We would like to thank the Reviewers for the very fruitful, constructive and helpful comments. In the following we address their concerns point by point.

REVIEWERS' COMMENTS:

Reviewer #1 (Remarks to the Author):

The authors appropriately responded to all the changes and incorporated important changes to the manuscript. This has improved the manuscript significantly, but there are still several issues that should be addressed before this manuscript should be accepted.

1. Readers may be confused by the use of the terms "study" and "experiment" in this manuscript. For example, in line 247, the reader may assume that experiment 2 and study 2 are the same, and they may be confused when experiment 3 is introduced in line 343. I suggest adding a short paragraph at the beginning of the Results section for study 2, explaining it is composed of experiments 2-4 that will be described in order.

We agree with the reviewer. We have now introduced a short paragraph at the beginning of the Results section for study 2, explaining it is composed of experiments 2-4 as follows (page 9):

“Study 2 – Cued memory reactivation during sleep

Study 2 was composed of three independent experiments (Exp. 2-4). We first examined the short-term effect of memory reactivation with different types of reminders during SWS (Exp. 2). Then, we looked at the long-term effect of memory reactivation with different types of reminders during SWS (Exp. 3). Finally, we tested whether the observed long-term effects depend on extended sleep or rather on the simple passage of time (Exp. 4). Note that in Exp. 2 and 4 the envisaged duration of the short sleep period was ~40 min as in Diekelmann et al. (2011)³⁸. Although some participants in Exp. 2 and 4 slept somewhat longer than that (Mean: 53.3 min, SD: 23.8, see methods section for details), we stick to *the term '40-min sleep' in the following for reasons of consistency.*”

2. The authors clarified what they mean by "a 40-minute nap", but this interpretation is non-trivial and may be misleading to authors who don't carefully

read the results section. In some cases, the phrasing is simply erroneous (e.g., line 380 states that there were 40 minutes of sleep, when in fact for some participants there were more than twice as many). I suggest one of the following: either define a term, such as "short-duration sleep", which will be introduced along with a short explanation and then used consistently; or rephrase as "at least 40 minutes" in key positions in the manuscript, along with a referral to the methods section. If the authors are set on keeping the current phrasing, I suggest adding a note at the beginning of the Results for study 2 explaining that 40-minute sleep periods are not limited to 40 minutes.

We agree with the reviewer suggestion. We decided to leave the name of the groups as they were but explaining in results and methods section that the short nap had not an exactly duration of 40 min. Furthermore, we added the symbol "~" before the term "40 min" throughout the manuscript.

Result section, page 9 (at the beginning of Study 2):

"Note that in Exp. 2 and 4 the envisaged duration of the short sleep period was ~40 min as in Diekelmann et al. (2011)³⁸. Although some participants in Exp. 2 and 4 slept somewhat longer than that (Mean: 53.3 min, SD: 23.8, see methods section for details), we stick to the term '40-min sleep' in the following for reasons of consistency."

Methods section, pages 32:

"For groups that received the reactivation during SWS, the experimenter had to wait 10 min after the subjects reached stable SWS before starting the reactivation. Thus, in some cases the experimenter had to wait longer until the participant reached stable SWS (amounting to longer sleep duration in some cases). Moreover, the reactivation was paused whenever signs of arousals or changes in sleep stage were detected, and resumed upon stable SWS was detected again, additionally adding some variance in individual sleep time."

3. In my previous point #8, I noted that there is some contradiction between the exclusion justification in line 751 and the fact that arousals were immediately followed by a temporary termination of cuing. This issue has not been resolved. Waking up involves an arousal, followed by a break in cuing. Assumingly, if the participants went back to sleep, cuing should start again and the participants should be included in the final sample. Perhaps the authors mean that some participants were not sufficiently cued because they did not go back into N3 within some timeframe. If so, they should mention what that timeframe was (in the current methods section, there is no upper boundary for the length of a sleep period) and rephrase the omission criteria. Alternatively, the authors may have excluded any participant that had even a single epoch of wake after sleep onset. This should be clarified and justified.

We thank the reviewer for this remark. We have now clarified the omission criteria in the methods section as follows (pages 32):

“For groups that received the reactivation during SWS, the experimenter had to wait 10 min after the subjects reached stable SWS before starting the reactivation. Thus, in some cases the experimenter had to wait longer until the participant reached stable SWS (amounting to longer sleep duration in some cases). Moreover, the reactivation was paused whenever signs of arousals or changes in sleep stage were detected, and resumed upon stable SWS was detected again, additionally adding some variance in individual sleep time. Furthermore, if participants did not reach stable SWS again after 90 min, the reactivation did not continue to prevent reactivation in the second sleep cycle, and the participant was excluded from the experiment.”

4. For the time-frequency analysis, equivalent NR time epochs were used as baseline. The authors should mention how those were chosen (e.g., were they chosen from N3? Were epochs within the first 10 minutes of stable SWS chosen or excluded like for the cued data? Were these epochs non-overlapping?)

We thank the reviewer for this comment. For the groups that did not receive any reactivation (NR groups), markers were assigned post-hoc starting after 10 min of stable SWS during the first 40 min of sleep. They could be in the same epoch but in non-overlapping segments. We moved this explanation from the paragraph on the reactivation session to Reactivation EEG data preprocessing and we added further information as follows (pages 34):

“The onset of each stimulus (sounds and cues) was automatically marked in the EEG recording during reactivation. For the groups that did not receive any reactivation (NR groups), markers were assigned post-hoc according to the same criteria (i.e. starting after 10 min of stable SWS during the first ~40 min of sleep, without overlapping the segments of reactivation).”

Reviewer #2 (Remarks to the Author):

Thanks for this good revision. You have answered all of my concerns, and I am now satisfied that the paper should be published.

Reviewer #3 (Remarks to the Author):

The authors addressed all my concerns. I have nothing more to add.